# TriC-Motion: Tri-Domain Causal Modeling Grounded Text-to-Motion Generation

**Yiyang Cao**[1][*], **Yunze Deng**[1][*][§], **Ziyu Lin**[2], **Bin Feng**[1][†], **Xinggang Wang**[1], **Wenyu Liu**[1],
**DanDan Zheng**[3][‡][§], **Jingdong Chen**[3][‡][§]

[1] School of Electronic Information and Communications, Huazhong University of Science
and Technology, Wuhan, China
[2] Whiting School of Engineering, Johns Hopkins University, Baltimore, Maryland, USA
[3] Ant Group, Beijing, China
{yiyangcao, yunzedeng, fengbin}@hust.edu.cn
{yuandan.zdd, jingdongchen}@antgroup.com

## ABSTRACT

Text-to-motion generation, a rapidly evolving field in computer vision, aims to produce realistic and text-aligned motion sequences. Current methods primarily focus on spatial-temporal modeling or independent frequency domain analysis, lacking a unified framework for joint optimization across spatial, temporal, and frequency domains. This limitation hinders the model's ability to leverage information from all domains simultaneously, leading to suboptimal generation quality. Additionally, in motion generation frameworks, motion-irrelevant cues caused by noise are often entangled with features that contribute positively to generation, thereby leading to motion distortion. To address these issues, we propose Tri-Domain Causal Text-to-Motion Generation (TriC-Motion), a novel diffusion-based framework integrating spatial-temporal-frequency-domain modeling with causal intervention. TriC-Motion includes three core modeling modules for domain-specific modeling, namely Temporal Motion Encoding, Spatial Topology Modeling, and Hybrid Frequency Analysis. After comprehensive modeling, a Score-guided Tri-domain Fusion module integrates valuable information from the triple domains, simultaneously ensuring temporal consistency, spatial topology, motion trends, and dynamics. Moreover, the Causality-based Counterfactual Motion Disentangler is meticulously designed to expose motion-irrelevant cues to eliminate noise, disentangling the real modeling contributions of each domain for superior generation. Extensive experimental results validate that TriC-Motion achieves superior performance compared to state-of-the-art methods, attaining an outstanding R@1 of 0.612 on the HumanML3D dataset. These results demonstrate its capability to generate high-fidelity, coherent, diverse, and text-aligned motion sequences. Code is available at: https://caoyiyang1105.github.io/TriC-Motion/.

## 1 INTRODUCTION

Text-driven human motion generation is rapidly emerging as a focal research area in computer vision, with broad potential impact across the film and game industry (Shuto et al., 2025), human–computer interaction (Wang et al., 2025; Sui et al., 2025), and embodied intelligence in robotics (Long et al., 2025). This task involves interpreting textual descriptions to produce smooth, physically plausible, and semantically coherent joint coordinate sequences, such as natural poses for walking, running, or jumping (Xue et al., 2025).

---

[*]Equal contribution.

[†]The first corresponding author.

[‡]The second corresponding authors.

[§]This work was supported by National Natural Science Foundation of China (No.62376102) and Ant Group.

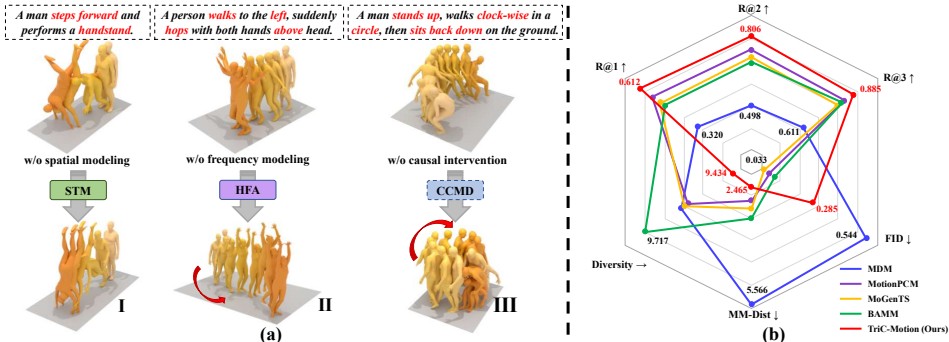

Figure 1: (a) Visual comparison of motion generated before and after spatial modeling/frequency modeling/causal intervention; (b) Quantitative comparison of different methods' performance on HumanML3D.

Recent text-to-motion methods can be broadly divided into diffusion-based (Tevet et al., 2022; Zhang et al., 2023b; 2024a; Zhou et al., 2024) and auto-regressive approaches (Zhang et al., 2023a; Pinyoanuntapong et al., 2024; Guo et al., 2024), primarily focusing on temporal modeling to depict dynamic evolution and ensure sequence consistency. Recent works (Yuan et al., 2024; Zhang et al., 2024b) extend to spatial-temporal joint modeling, further enhancing motion realism and topological consistency while improving motion quality. Typically, Spatio-Temporal Graph Diffusion (Liu et al., 2023) encodes local joint topologies via graph convolutions and characterizes dynamic evolution through 1-D temporal convolutions, while HiSTF Mamba (Zhan et al., 2025) captures short- and long-range spatial-temporal clues using bidirectional Mamba. However, generating motions with high or complex dynamics still remains challenging.

Given the successful application of spectral analysis in various fields (Hyun et al., 2023; Chen et al., 2024a; Li et al., 2024b; Kim et al., 2024), some motion generation methods (Xu & Chen, 2024; Li et al., 2024a; Wan et al., 2023) address the aforementioned issues by independently analyzing low-frequency and high-frequency signals. For human motion, low-frequency components capture global evolution for smoothness (*i.e.*, coarse motion trends), while high-frequency components depict subtle joint dynamics for richness (*i.e.*, fine motion details) (Li et al., 2024a). Jointly attending to these complementary bands promotes generations that are both coherent and richly detailed.

Despite significant progress, a unified motion generation framework integrating spatial, temporal, and frequency domains remains unexplored. Fig. 1(a)I highlights that missing spatial modeling leads to unrealistic joint topology, while Fig. 1(a)II demonstrates that high- and low-frequency modeling ensures overall trends and fine dynamic details. Therefore, we argue that integrating all three domains into one framework fully utilizes complementary information, enabling comprehensive motion representations for high-quality generation. This approach has proven effective in various tasks such as target detection (Duan et al., 2024; Huang et al., 2025; Duan et al., 2025), classification (Liu et al., 2021), and image decoding (Cao et al., 2024), highlighting its potential for motion generation.

Another potential issue is that motion-irrelevant cues caused by noise during motion generation modeling are often entangled within the features, thereby leading to ineffective modeling and degrading fidelity. In a naive multi-domain architecture, this issue even worsens as noise accumulates across domains. Inspired by causal intervention (Pearl et al., 2016; Yang et al., 2023; Xiong et al., 2024), we identify this problem as the model's inability to distinguish beneficial *factual features* from motion-irrelevant *counterfactual features*. To address this, we employ a structural causal model within our tri-domain architecture (Sec. 2) to extract beneficial causal contributions, disentangle and eliminate motion-irrelevant cues, thereby focusing on valuable complementary cross-domain information. As shown in Fig. 1(a)III, such intervention effectively mitigates quality degradation caused by multi-domain noise accumulation, ultimately enhancing motion fidelity.

Therefore, we propose **Tri**-Domain **C**ausal Text-to-**Motion** Generation (TriC-Motion), a novel framework integrating causal intervention with tri-domain modeling for motion generation. Built on MDM (Tevet et al., 2022), it employs multiple TriC-Motion Denoiser Blocks, each including three core modules: Temporal Motion Encoding (TME), Spatial Topology Modeling (STM), and Hybrid Frequency Analysis (HFA) modules. Subsequently, the **Score-guided Tri-domain Fusion (S-Fus)** module efficiently integrates tri-domain information, ensuring temporal consistency, spatial topol-

ogy, accurate overall motion trends, and fine-grained dynamics. Additionally, during training, the Causality-based Counterfactual Motion Disentangler (CCMD) is applied in each block to disentangle motion-irrelevant cues, guiding tri-domain modeling toward key motion information. To the best of our knowledge, TriC-Motion is the first method to introduce causal intervention into motion generation. Compared to previous methods, it captures essential tri-domain features, enabling higher-quality motion generation. As shown in Fig. 1(b), TriC-Motion achieves 0.612 of R1-Precision (R@1) and sets new state-of-the-art performance across most metrics on HumanML3D (Guo et al., 2022), with visualizations showing improved fidelity, diversity, and semantic consistency.

Our main contributions can be summarized as follows: (1) We propose a novel motion generation framework that unifies spatial-temporal-frequency modeling within a diffusion-based denoising architecture. The collaborative integration of information from the three domains enables the model to capture temporal dynamics, spatial topology, and multi-granularity frequency characteristics, thereby improving generation quality and fidelity. (2) We pioneer the introduction of causal intervention into motion generation, designing an innovative Causality-based Counterfactual Motion Disentangler module. This module effectively removes redundant information and noise in tri-domain modeling, stabilizes the denoising process, and guides each domain to focus on key motion features for better generation. (3) We conduct comprehensive evaluations on HumanML3D and SnapMoGen, achieving state-of-the-art results on most metrics. Extensive ablation studies further confirm the effectiveness of each domain branch and the causal intervention design.

## 2    RELATED WORK

### 2.1    SPATIAL-TEMPORAL-FREQUENCY MODELING IN MOTION GENERATION

Temporal modeling (Zhang et al., 2024a; Tevet et al., 2022; Zhang et al., 2024b; 2023a; Guo et al., 2024; Pinyoanuntapong et al., 2024) has been the dominant paradigm in text-to-motion generation. Spatial priors further enhance plausibility, such as graph-based spatio-temporal convolutions (Liu et al., 2023) and joint-token attention (Yuan et al., 2024). Frequency modeling complements these by leveraging low- and high-frequency components for global trends and fine details (Wan et al., 2023; Li et al., 2024a). However, a unified framework that jointly optimizes temporal, spatial, and frequency cues remains underexplored, hindering coherent, physically plausible, and high-quality motion generation.

### 2.2    CAUSALITY IN COMPUTER VISION

Causality has become an increasingly powerful tool in computer vision: (Yang et al., 2021) mitigate confounding in vision–language alignment via causal attention; (Chen et al., 2023a) address domain shift through meta-causal learning; and, for gait recognition, counterfactual interventions are adopted to suppress non-identity factors(*e.g.*, appearance, load-carrying)(Xiong et al., 2024; Dou et al., 2023). Yet generative modeling, especially text-to-motion, remains underexplored. TriC-Motion adopts an SCM-inspired view of tri-domain features as mixtures of intrinsic signals and confounders and performs interventions during diffusion to preserve causal contributions and remove motion-irrelevant information.

## 3    METHOD

### 3.1    PRELIMINARY

**Causal Learning.** Causal inference and causal intervention are the core concepts of causal learning, with the former identifying cause-and-effect relationships and the latter disrupting original causal connections (Pearl et al., 2016). Inspired by Structured Causal Models (Zhang, 1994), we present Fig. 2 to better illustrate how TriC-Motion integrates causal learning into motion generation. Specifically, the $j$-th layer of TriC-Motion Denoiser Block extracts domain-specific features $F_j^i (i \in \{temp, spa, freq\})$ through carefully designed

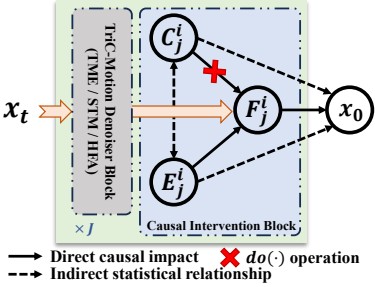

Figure 2: Structured Casual Model in TriC-Motion.

modeling, which are typically entangled results of intrinsic motion characteristics $E_j^i$ and confounders $C_j^i$ (e.g., motion-irrelevant cues) that adversely affect generation. These features are then used to generate the motion sequence $x_0$. Ideally, $F_j^i$ should consist solely of $E_j^i$, *i.e.*, $E_j^i \rightarrow F_j^i \rightarrow x_0$, where $\rightarrow$ denotes direct causal impact. However, in practice, the motion feature modeling process invariably includes motion-irrelevant cues, which couple with $E_j^i$, leading to $(C_j^i, E_j^i) \rightarrow F_j^i \rightarrow x_0$. Since directly removing $C_j^i$ is challenging, we design the CCMD module to perform causal intervention $do(\cdot)$, isolating the causal impact of $C_j^i$ on $F_j^i$. This operation ensures that generated motions are influenced primarily by the intrinsic motion characteristics rather than confounders, thereby enhancing the overall quality.

## 3.2 PIPELINE

The overview of TriC-Motion is illustrated in Fig. 3(b). Our framework can be divided into two key parts: TriC-Motion Denoiser Blocks and Causal Intervention Blocks. The former are dedicated to capturing the tri-domain characteristics of motion sequences, while the latter disentangle and eliminate motion-irrelevant cues from each domain.

TriC-Motion builds upon MDM (Tevet et al., 2022)'s diffusion and sampling processes, and its sampling process and network architecture are illustrated in Fig. 3. Following (Yuan et al., 2024), the motion sequence is preprocessed from a 1D temporal structure to a 2D spatial-temporal structure to facilitate subsequent modeling. Then, the sequence is downsampled temporally by a factor of $s$ and projected into the feature dimension via a linear layer, forming $X \in \mathbb{R}^{N \times M \times D}$, where $N$, $M$, and $D$ denote downsampled frames, number of joints, and feature dimensionality, respectively. The diffusion timestep $t$ is omitted for simplicity. To incorporate spatial-temporal position information, the original 1D sinusoidal positional encoding is replaced by its 2D variant (Wang & Liu, 2021), which is then added to the motion sequence. Besides, the pretrained text encoder DistilBERT (Sanh et al., 2019) encodes the provided prompt, generating sentence-level features $CLS$ and word-level features $\tau$. The diffusion timestep $t$ is projected to the feature dimension via a feed-forward network and concatenated with $CLS$ and motion features, yielding $X \in \mathbb{R}^{(N+2) \times M \times D}$.

The TriC-Motion Denoiser stacks $J$ layers of identical blocks. At the $j$-th layer and timestep $t$, motion features $X_j$ are processed in parallel across temporal, spatial, and frequency domains by three modules (TME, STM, HFA), yielding domain-specific features $F_j^i$, where $i \in \{temp, spa, freq\}$ represents the domain index. Following this, the S-Fus module integrates $F_j^i$ by leveraging semantic information and motion-specific properties from all domains, producing the fused feature $Y_j$. Finally, semantic information is injected via cross-attention in the **Textual Information Injection (TIJ)** module, where word-level features $\tau$ act as keys and values, and fused motion features serve as queries. The complete process of the tri-domain modeling over $J$ blocks can be formulated as:

$$\begin{cases} \text{TIJ}(X_j, \tau) = \text{CrossAttention}(X_j, \tau, \tau), \\ \hat{X} = [\text{TIJ}(\text{S-Fus}(\text{TME}(X_j), \text{STM}(X_j), \text{HFA}(X_j), CLS), \tau)]||_{j=1}^J \end{cases} \quad (1)$$

where "$||_{j=1}^J$" represents stacking $J$ times, and S-Fus$(\cdot)$ denotes the process of S-Fus module. Furthermore, to eliminate motion-irrelevant cues, CCMD is applied to the features $F_j^i$ obtained after domain-specific modeling. This block forces tri-domain modeling to disentangle essential components from motion-irrelevant parts in generation features by leveraging causal loss $\mathcal{L}_{fcf,j}$ to guide gradient-based optimization. Importantly, CCMD is utilized exclusively during training.

## 3.3 TRI-DOMAIN MODELING FOR GENERATION

In this subsection, we introduce the five core components of the TriC-Motion Denoiser Block, *i.e.* TME, STM, HFA, S-Fus, and TIJ, as their details are shown in Fig. 4.

**Temporal Motion Encoding.** TME leverages a vanilla TransformerEncoderLayer that attends over motion frames along the temporal dimension, explicitly capturing both short- and long-range dependencies for temporally coherent motion modeling. The output of TME can be expressed as:

$$F_j^{temp} = \text{TransformerEncoderLayer}(X_j) \quad (2)$$

**Spatial Topology Modeling.** The skeleton of human motion sequence at each frame can be naturally represented as a graph, with joints as nodes and body segments as edges. To efficiently capture local

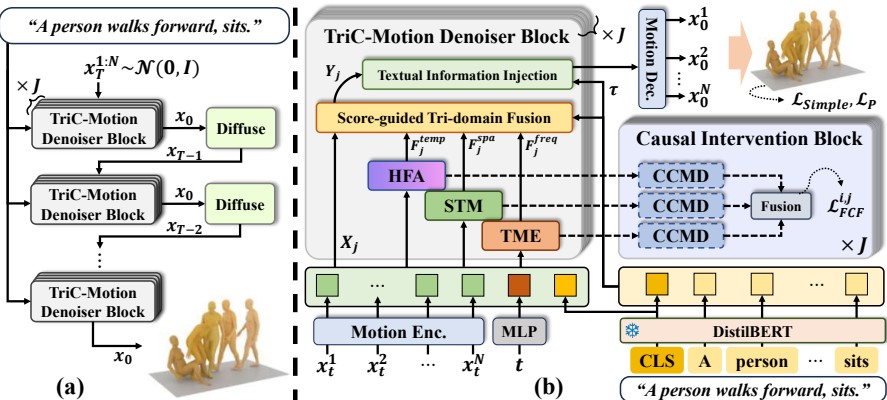

Figure 3: Overview of TriC-Motion. (a) Sampling process with stacked TriC-Motion Denoiser Blocks. (b) Overall architecture of the TriC-Motion framework.

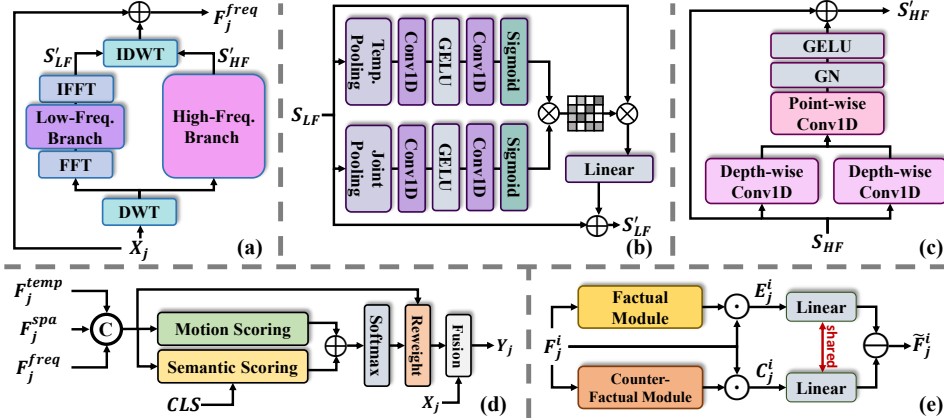

Figure 4: Detailed architectures of TriC-Motion main components. (a) HFA with DWT/FFT decomposition; (b) Low-frequency branch network in HFA; (c) High-frequency branch network in HFA; (d) S-Fus with motion and semantic scoring; (e) Details of CCMD.

topology and inter-joint dependencies, the STM module is built upon a graph convolutional network (GCN) (Kipf, 2016), enhancing the naturalness and physical plausibility of generated motion. In STM, a 3-layer GCN network is employed to model the joint dimension of $X_j$:

$$F_j^{spa} = X_j + [\text{LN}(\text{GELU}(\text{GCN}(X_j)))]||^3 \tag{3}$$

where "$||^3$" represents stacking 3 times, and $\text{LN}(\cdot)$ denotes layer normalization.

**Hybrid Frequency Analysis.** Human motions exhibit distinct frequency characteristics: low-frequency components capture global motion trends, while high-frequency components reflect instantaneous changes and fine-grained details. Inspired by this, we propose the Hybrid Frequency Analysis (HFA) module (Fig. 4(b)), which combines Discrete Wavelet Transform (DWT) and Fast Fourier Transform (FFT) to leverage their synergy, capturing localized temporal-frequency dynamics via wavelet decomposition and global frequency patterns via Fourier analysis (Kiruluta & Lemos, 2025). Specifically, the motion sequence $X_j$ is decomposed into low-frequency sub-band $\hat{S}LF$ and high-frequency sub-band $S_{HF}$ via DWT, followed by FFT applied to $\hat{S}_{LF}$:

$$(\hat{S}_{LF}, S_{HF}) = \text{DWT}(X_j), \text{and } S_{LF} = \text{FFT}(\hat{S}_{LF}). \tag{4}$$

Utilizing the distinct characteristics of low- and high-frequency components, specialized branches are designed for targeted analysis. As shown in Fig. 4(c), the low-frequency branch adaptively attends to key spatial-temporal regions to highlight critical motion patterns. Two parallel convolution networks are respectively applied to $S_{LF}$ across temporal and joint dimensions to extract global

contextual information. This information is further used to generate a 2D spatial-temporal attention matrix to optimize the low-frequency signal, and the whole process can be expressed as:

$$S'_{LF} = S_{LF} + \text{Linear}(S_{LF} \otimes (w_t \otimes w_s)). \tag{5}$$

The high-frequency branch employs a lightweight convolutional architecture to capture fine-grained local motion details. As shown in Fig. 4(d), it utilizes two 1D depth-wise convolutions $f^d(\cdot)$ to extract spatial and temporal spectral information, which are then integrated through a 1D point-wise convolution $f^p(\cdot)$ to enhance the high-frequency signal $S_{HF}$:

$$S'_{HF} = S_{HF} + \text{GELU}(\text{GN}(f^p(f^d(S_{HF})))), \tag{6}$$

where $\text{GN}(\cdot)$ is the Group Normalization (Wu & He, 2018). After conducting analysis on the low-frequency and high-frequency branches, the enhanced features $S'_{LF}$ and $S'_{HF}$ are obtained. Finally, $\text{IFFT}(\cdot)$ and $\text{IDWT}(\cdot)$ are applied to merge and transform them back to the spatial-temporal domain.

**Score-guided Tri-domain Fusion.** As shown in Fig. 4(d), the S-Fus module employs a dual-branch scoring framework to integrate tri-domain features using global semantic context and motion features. It consists of two scoring branches: Motion Scoring and Semantic Scoring. The former produces motion logits $logits_{mot}$ to capture multi-domain correlations, while the latter leverages the global semantic token $CLS$ to generate semantic logits $logits_{sem}$, ensuring semantic consistency during fusion. Afterward, the two logits are normalized via softmax to obtain domain-specific weights $\alpha_i$, which are used for weighted fusion of tri-domain features $Y_j$. The above process can be expressed as Eq. 7, where $i \in \{temp, spa, freq\}$. Here, $F_j^{tri}$ denotes $\text{CAT}(F_j^{temp}, F_j^{spa}, F_j^{freq})$. Both $f_{mot}(\cdot)$ and $f_{sem}(\cdot)$ share the same feed-forward architecture, implemented as $\text{Linear}(\text{GELU}(\text{Linear}(\cdot)))$.

$$\begin{cases} logits_{mot}^i = f_{mot}(F_j^{tri}), \text{and } logits_{sem}^i = f_{sem}(\text{CAT}(F_j^i, CLS)) \\ \alpha_i = \text{Softmax}(logits_{mot}^i + logits_{sem}^i) \\ Y_j = \text{Linear}(\text{CAT}(X_j, \sum_i \alpha_i F_j^i)). \end{cases} \tag{7}$$

## 3.4 CAUSALITY-BASED COUNTERFACTUAL MOTION DISENTANGLE

The CCMD module is applied to each layer of TriC-Motion Denoiser to disentangle motion-irrelevant cues from beneficial causal contributions in tri-domain modeling. As shown in Fig.4, it includes two key components: the Factual and Counterfactual Modules. The two modules adopt a lightweight symmetric architecture to efficiently extract causal contributions $E_j^i$ and confounders $C_j^i$ from $F_j^i$. Taking Factual Module as an example, the detailed network can be expressed as:

$$\begin{cases} \omega = \text{Sigmoid}(\text{Linear}(\text{ReLU}(\text{Linear}(\text{Pool}(F_j^i))))) \in \mathbb{R}^{1 \times 1 \times D} \\ E_j^i = \text{Linear}(\omega F_j^i) \odot F_j^i \end{cases}, \tag{8}$$

where $\text{Pool}(\cdot) = \text{AvgPool}(\cdot) + \text{MaxPool}(\cdot)$, and the counterfactual features $C_j^i$ can be obtained similarly. Next, we eliminate confounders by performing supervised causal intervention, calculated as $\tilde{F}_j^i = W_{do}E_j^i - W_{do}C_j^i$, where $W_{do}$ is implemented via linear mappings.

## 3.5 LOSS FUNCTION

Our model uses $\mathcal{L}_{\text{simple}}$ as the primary training objective for motion generation, where the denoising process predicts the clean motion sequence $\hat{x}_0 = f(x_t, t, c)$ from its noisy counterpart:

$$\mathcal{L}_{\text{simple}} = \mathbb{E}_{x_0 \sim q(x_0|c), t \sim [1,T]} \left[ \|x_0 - f(x_t, t, c)\|^2 \right] \tag{9}$$

where $c$ and $x_0$ denote the prompt condition and the ground truth motion sequence, respectively. In addition, we introduce the Factual and Counterfactual Loss $\mathcal{L}_{fcf,j}$ to constrain each CCMD module. In the $j$-th layer, the CCMD module produces tri-domain $\tilde{F}_j$ by concatenating features from three domains. This loss enforces the results of tri-domain causal intervention to approach the ground truth, thereby supervising CCMD to eliminate motion-irrelevant cues:

$$\mathcal{L}_{fcf} = \sum_{j=1}^{J} w_j \mathcal{L}_{fcf,j} = \sum_{j=1}^{J} w_j \mathcal{L}_{MSE}(TDE_j, x_0) \tag{10}$$

Table 1: Quantitative results on HumanML3D. The right arrow → means the closer to real motion the better. Each experiment is repeated 20 times, with average results and 95% confidence intervals (±) reported. The best result is highlighted in **bold**, and the second-best is underlined.

| Methods | R-Precision ↑ | | | FID ↓ | MM Dist ↓ | Diversity → |
|---|---|---|---|---|---|---|
| | Top 1 | Top 2 | Top 3 | | | |
| GT | $0.511^{\pm.003}$ | $0.703^{\pm.003}$ | $0.797^{\pm.002}$ | $0.002^{\pm.000}$ | $2.974^{\pm.008}$ | $9.503^{\pm.065}$ |
| MDM (Tevet et al., 2022) | $0.320^{\pm.005}$ | $0.498^{\pm.004}$ | $0.611^{\pm.007}$ | $0.544^{\pm.044}$ | $5.566^{\pm.027}$ | $\mathbf{9.559}^{\pm.086}$ |
| M2DM (Kong et al., 2023) | $0.497^{\pm.003}$ | $0.682^{\pm.002}$ | $0.763^{\pm.003}$ | $0.352^{\pm.005}$ | $3.134^{\pm.010}$ | $9.926^{\pm.073}$ |
| CoMo (Huang et al., 2024b) | $0.502^{\pm.002}$ | $0.692^{\pm.007}$ | $0.790^{\pm.002}$ | $0.262^{\pm.004}$ | $3.032^{\pm.015}$ | $9.936^{\pm.066}$ |
| StableMoFusion (Huang et al., 2024a) | $0.553^{\pm.003}$ | $0.748^{\pm.002}$ | $0.841^{\pm.002}$ | $0.098^{\pm.003}$ | - | $9.748^{\pm.092}$ |
| BAMM (Pinyoanuntapong et al., 2024) | $0.525^{\pm.002}$ | $0.720^{\pm.003}$ | $0.814^{\pm.003}$ | $0.055^{\pm.002}$ | $2.919^{\pm.008}$ | $9.717^{\pm.089}$ |
| MoGenTS (Yuan et al., 2024) | $0.529^{\pm.003}$ | $0.719^{\pm.002}$ | $0.812^{\pm.002}$ | $\underline{0.033}^{\pm.001}$ | $2.867^{\pm.006}$ | $\underline{9.570}^{\pm.077}$ |
| MoMask (Guo et al., 2024) | $0.521^{\pm.002}$ | $0.713^{\pm.002}$ | $0.807^{\pm.002}$ | $0.045^{\pm.002}$ | $2.958^{\pm.008}$ | - |
| MotionPCM (Jiang et al., 2025) | $0.560^{\pm.002}$ | $0.754^{\pm.002}$ | $0.844^{\pm.002}$ | $0.040^{\pm.003}$ | $2.719^{\pm.008}$ | $9.632^{\pm.089}$ |
| MARDM (Meng et al., 2024) | $0.500^{\pm.004}$ | $0.659^{\pm.003}$ | $0.795^{\pm.003}$ | $0.114^{\pm.007}$ | $3.270^{\pm.009}$ | - |
| MotionLCM-v2 (Dai et al., 2024) | $0.551^{\pm.003}$ | $0.745^{\pm.002}$ | $0.836^{\pm.002}$ | $0.049^{\pm.003}$ | $2.765^{\pm.009}$ | $9.584^{\pm.066}$ |
| LaMP (Li et al., 2024c) | $0.557^{\pm.003}$ | $0.751^{\pm.002}$ | $0.843^{\pm.001}$ | $\mathbf{0.032}^{\pm.002}$ | $2.759^{\pm.007}$ | $9.571^{\pm.069}$ |
| GMMotion (Tu et al.) | $0.572^{\pm.003}$ | $0.761^{\pm.003}$ | $0.852^{\pm.001}$ | $0.086^{\pm.003}$ | $2.743^{\pm.008}$ | $9.792^{\pm.085}$ |
| SALAD (Hong et al., 2025) | $0.581^{\pm.003}$ | $0.769^{\pm.003}$ | $0.857^{\pm.002}$ | $0.076^{\pm.002}$ | $2.649^{\pm.009}$ | $9.696^{\pm.096}$ |
| **TriC-Motion (Ours, base)** | $\underline{0.607}^{\pm.005}$ | $\underline{0.800}^{\pm.004}$ | $\mathbf{0.886}^{\pm.004}$ | $0.347^{\pm.031}$ | $\mathbf{2.463}^{\pm.012}$ | $9.428^{\pm.085}$ |
| **TriC-Motion (Ours, large)** | $\mathbf{0.612}^{\pm.006}$ | $\mathbf{0.806}^{\pm.005}$ | $\underline{0.885}^{\pm.004}$ | $0.285^{\pm.042}$ | $\underline{2.465}^{\pm.017}$ | $9.434^{\pm.089}$ |

Table 2: Quantitative results on SnapMoGen test dataset.

| Methods | R-Precision ↑ | | | FID ↓ | CLIP Score ↑ | Diversity → |
|---|---|---|---|---|---|---|
| | Top 1 | Top 2 | Top 3 | | | |
| GT | $0.940^{\pm.001}$ | $0.976^{\pm.001}$ | $0.985^{\pm.001}$ | $0.001^{\pm.000}$ | $0.837^{\pm.000}$ | $19.756^{\pm.047}$ |
| MDM (Tevet et al., 2022) | $0.503^{\pm.002}$ | $0.653^{\pm.002}$ | $0.727^{\pm.002}$ | $57.783^{\pm.092}$ | $0.481^{\pm.001}$ | - |
| T2M-GPT (Zhang et al., 2023a) | $0.618^{\pm.002}$ | $0.773^{\pm.002}$ | $0.812^{\pm.002}$ | $32.629^{\pm.087}$ | $0.573^{\pm.011}$ | - |
| StableMoFusion (Huang et al., 2024a) | $0.679^{\pm.002}$ | $0.823^{\pm.002}$ | $0.888^{\pm.002}$ | $27.801^{\pm.063}$ | $0.605^{\pm.001}$ | - |
| MARDM (Meng et al., 2024) | $0.659^{\pm.002}$ | $0.812^{\pm.002}$ | $0.860^{\pm.002}$ | $26.878^{\pm.131}$ | $0.602^{\pm.001}$ | - |
| MoMask (Guo et al., 2024) | $0.777^{\pm.002}$ | $0.888^{\pm.002}$ | $0.927^{\pm.002}$ | $\underline{17.404}^{\pm.051}$ | $0.664^{\pm.001}$ | - |
| MoMask++ (Guo et al., 2025) | $\underline{0.802}^{\pm.001}$ | $\underline{0.905}^{\pm.003}$ | $\underline{0.938}^{\pm.001}$ | $\mathbf{15.060}^{\pm.065}$ | $\mathbf{0.685}^{\pm.001}$ | $\underline{19.970}^{\pm.048}$ |
| **TriC-Motion (Ours)** | $\mathbf{0.907}^{\pm.002}$ | $\mathbf{0.969}^{\pm.001}$ | $\mathbf{0.985}^{\pm.001}$ | $26.346^{\pm.073}$ | $\underline{0.675}^{\pm.001}$ | $\mathbf{19.831}^{\pm.042}$ |

where $w_j$ represents the weight for each layer, with values $\{0.1, 0.2, 0.3, 0.4\}$ when J = 4. To improve the perceptual quality of generated motions, we augment $\mathcal{L}_{\text{simple}}$ with a perceptual loss $\mathcal{L}_p$, following the common practice in image processing (Johnson et al., 2016). It operates in a learned feature space to better capture semantic motion characteristics. Using a pre-trained motion encoder (Guo et al., 2022; 2025) $\mathcal{E}$, we extract feature representations from generated motion $\hat{x}_0$ and ground truth $x_0$. The perceptual loss is defined as L2-norm of their feature difference: $\mathcal{L}_p = \|\mathcal{E}(\hat{x}_0) - \mathcal{E}(x_0)\|_2^2$, ensuring the generated motions are perceptually indistinguishable from real ones, enhancing overall quality. The total loss is defined in Eq. 11, with $\lambda_{fcf} = 1$ and $\lambda_p = 10$.

$$\mathcal{L} = \mathcal{L}_{\text{simple}} + \lambda_{fcf}\mathcal{L}_{fcf} + \lambda_p\mathcal{L}_p \tag{11}$$

## 4 EXPERIMENTS

### 4.1 EXPERIMENTAL SETUP

**Datasets.** We evaluate TriC-Motion on two large-scale motion–language benchmarks: HumanML3D (Guo et al., 2022) and SnapMoGen (Guo et al., 2025). HumanML3D contains 14,616 motion sequences and 44,970 texts aggregated from AMASS (Mahmood et al., 2019) and Human-Act12 (Guo et al., 2020), covering diverse activities (*e.g.*, walking, acrobatics). SnapMoGen comprises 20,450 motion clips, each paired with six textual descriptions (two manually annotated and four LLM-augmented), totaling 122,565 descriptions with an average length of 48 words.

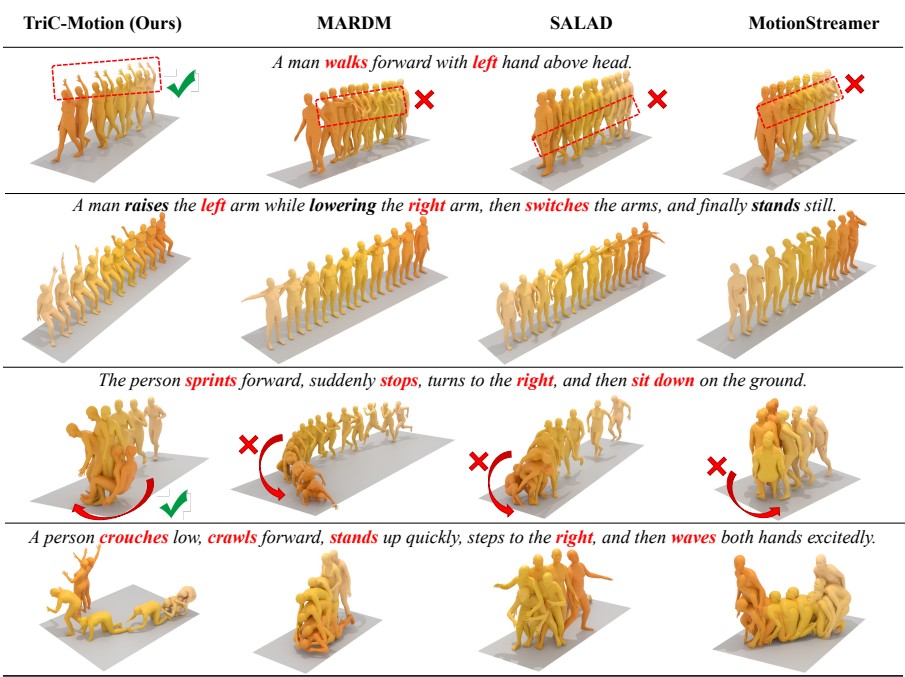

Figure 5: Qualitative comparisons on HumanML3D dataset.

**Evaluation Metrics.** We adopt the same evaluation metrics established in (Guo et al., 2022; 2025): (1) *Fréchet Inception Distance (FID)*, which measures the distribution distance of the generated and ground-truth motions; (2) *R-Precision (R@1, R@2, &R@3)*, *Multimodal Distance (MM-Dist)* and *CLIP Score*, which evaluate the semantic alignment between the input text and the generated motions; (3) *Diversity*, which measures the variability and richness of the generations.

**Implementation Details.** The diffusion timesteps are set to 50 with a cosine variance schedule for $\beta_t$. The number of denoiser blocks $J$ and feature dimension $D$ are set to 4 and 256, respectively. Two variants of TriC-Motion are implemented, with the temporal downsampling factor $s$ set to 7 for the base model and 4 for the large model. Classifier-free guidance is applied with scale $g = 4.0$. The model is trained on two NVIDIA RTX 3090 GPUs with a batch size of 64. AdamW is employed as the optimizer with learning rate $1 \times 10^{-4}$. Total training iterations are 650K for HumanML3D and 250K for SnapMoGen. Additionally, for both datasets, we apply the same approach to reconstruct the pose sequences, obtaining a spatial-temporal 2D format of dimension $M \times N \times 12$.

## 4.2 Experimental Results

**Quantitative Comparison.** We compare our model with several existing state-of-the-art methods on both HumanML3D and SnapMoGen datasets, with the results summarized in Tab. 1 and Tab. 2, respectively. Our method significantly outperforms all previous approaches in terms of R-Precision and MM-Dist, surpassing the second-best method, SALAD (Hong et al., 2025), by +0.031 in R@1 and -0.184 in MM-Dist, respectively, highlighting substantial improvement of TriC-Motion in text–motion alignment. Furthermore, when confronted with more complex and lengthy prompts in SnapMoGen, our method maintains robust text consistency, markedly outperforming the second-best method, MoMask++. With respect to FID, our approach achieves a substantial reduction relative to the baseline on both datasets, indicating improved motion fidelity.

**Qualitative Comparison.** Fig. 5 compares our method with MARDM (Meng et al., 2024), SALAD (Hong et al., 2025), and MotionStreamer (Xiao et al., 2025). Across all examples, the competing methods struggle to accurately follow fine-grained textual instructions. MARDM fails to follow multi-step instructions and produces low-quality actions (*e.g.*, "*sit down*" and "*crawl*") with unrealistic body structures. SALAD and MotionStreamer improve overall motion smoothness and visual quality, yet both still struggle to adhere to fine-grained cues and have difficulty producing complete and coherent sequences for long, compositional prompts. For instance, key actions such

Table 3: Ablation study of the proposed modules in TriC-Motion on HumanML3D test dataset, as well as the analysis of HFA. *"2D rep"* denotes the spatio-temporal 2D motion representation with dimensions M × N × 12. *"low"* and *"high"* denote low- and high-frequncy modeling in HFA. *"w/o joint"* means removing the joint-wise frequency branch while retaining the temporal one.

| Method | R-Precision ↑ | | | FID ↓ | MM Dist ↓ | Diversity → |
|---|---|---|---|---|---|---|
| | Top 1 | Top 2 | Top 3 | | | |
| GT | $0.511^{\pm.003}$ | $0.703^{\pm.003}$ | $0.797^{\pm.002}$ | $0.002^{\pm.000}$ | $2.974^{\pm.008}$ | $9.503^{\pm.065}$ |
| Baseline | $0.320^{\pm.005}$ | $0.498^{\pm.004}$ | $0.611^{\pm.007}$ | $0.544^{\pm.004}$ | $5.566^{\pm.027}$ | $\underline{9.559}^{\pm.086}$ |
| TME (*2D rep*) | $0.470^{\pm.006}$ | $0.671^{\pm.006}$ | $0.777^{\pm.007}$ | $2.110^{\pm.071}$ | $3.293^{\pm.023}$ | $8.016^{\pm.094}$ |
| TME + STM | $0.570^{\pm.004}$ | $0.771^{\pm.005}$ | $0.859^{\pm.005}$ | $0.611^{\pm.065}$ | $2.617^{\pm.018}$ | $9.157^{\pm.120}$ |
| TME + HFA | $0.583^{\pm.007}$ | $0.778^{\pm.004}$ | $\underline{0.869}^{\pm.004}$ | $\underline{0.374}^{\pm.015}$ | $2.576^{\pm.017}$ | $\mathbf{9.535}^{\pm.072}$ |
| TME + STM + HFA | $\underline{0.592}^{\pm.005}$ | $\underline{0.780}^{\pm.006}$ | $0.867^{\pm.005}$ | $0.383^{\pm.028}$ | $\underline{2.564}^{\pm.020}$ | $9.679^{\pm.082}$ |
| TME + STM + HFA + S-Fus **(Ours)** | $\mathbf{0.607}^{\pm.005}$ | $\mathbf{0.800}^{\pm.004}$ | $\mathbf{0.886}^{\pm.004}$ | $\mathbf{0.347}^{\pm.031}$ | $\mathbf{2.463}^{\pm.012}$ | $9.428^{\pm.085}$ |
| *w/o* HFA | $0.572^{\pm.004}$ | $0.765^{\pm.004}$ | $0.863^{\pm.004}$ | $0.593^{\pm.052}$ | $2.599^{\pm.022}$ | $9.243^{\pm.083}$ |
| *w/o* FFT | $0.595^{\pm.006}$ | $0.790^{\pm.006}$ | $0.877^{\pm.004}$ | $\underline{0.405}^{\pm.050}$ | $2.518^{\pm.018}$ | $\underline{9.335}^{\pm.095}$ |
| *w/o* joint | $0.599^{\pm.008}$ | $0.793^{\pm.005}$ | $0.881^{\pm.004}$ | $0.418^{\pm.035}$ | $2.527^{\pm.018}$ | $9.188^{\pm.107}$ |
| *w/o* High-band | $\underline{0.602}^{\pm.006}$ | $\underline{0.799}^{\pm.005}$ | $\underline{0.885}^{\pm.005}$ | $0.504^{\pm.048}$ | $\underline{2.494}^{\pm.015}$ | $9.029^{\pm.080}$ |
| *w/* low + high **(Ours)** | $\mathbf{0.607}^{\pm.005}$ | $\mathbf{0.800}^{\pm.004}$ | $\mathbf{0.886}^{\pm.004}$ | $\mathbf{0.347}^{\pm.031}$ | $\mathbf{2.463}^{\pm.012}$ | $\mathbf{9.428}^{\pm.085}$ |

Table 4: Ablation experiment of CCMD (upper half) and the layer-wise loss weights of $\mathcal{L}_{fcf}$ (lower half) on HumanML3D. *"pre"* means that applying causal intervention before S-Fus, whereas *"post"* employs it afterward. *"temp"* denotes applying CCMD only to the temporal domain, while *"temp+spa"* denotes applying CCMD jointly to the temporal and spatial domains. $\{w_1, w_2, w_3, w_4\}$ denotes the weights for $\mathcal{L}_{fcf}$ when $J = 4$.

| Method | R-Precision ↑ | | | FID ↓ | MM Dist ↓ | Diversity → |
|---|---|---|---|---|---|---|
| | Top 1 | Top 2 | Top 3 | | | |
| GT | $0.511^{\pm.003}$ | $0.703^{\pm.003}$ | $0.797^{\pm.002}$ | $0.002^{\pm.000}$ | $2.974^{\pm.008}$ | $9.503^{\pm.065}$ |
| *w/o* CCMD | $0.568^{\pm.007}$ | $0.767^{\pm.007}$ | $0.859^{\pm.006}$ | $0.561^{\pm.060}$ | $2.624^{\pm.023}$ | $9.187^{\pm.088}$ |
| *w/* CCMD (*post*) | $\underline{0.604}^{\pm.089}$ | $\underline{0.798}^{\pm.007}$ | $\underline{0.880}^{\pm.005}$ | $\mathbf{0.328}^{\pm.027}$ | $\underline{2.512}^{\pm.018}$ | $\mathbf{9.482}^{\pm.068}$ |
| *w/* CCMD (*temp*) | $0.580^{\pm.008}$ | $0.773^{\pm.008}$ | $0.859^{\pm.007}$ | $0.514^{\pm.049}$ | $2.617^{\pm.024}$ | $\underline{9.467}^{\pm.092}$ |
| *w/* CCMD (*temp + spa*) | $0.602^{\pm.005}$ | $0.792^{\pm.006}$ | $0.875^{\pm.006}$ | $\underline{0.329}^{\pm.034}$ | $2.525^{\pm.019}$ | $9.345^{\pm.094}$ |
| *w/* CCMD (*temp + spa + freq*, **Ours**) | $\mathbf{0.607}^{\pm.005}$ | $\mathbf{0.800}^{\pm.004}$ | $\mathbf{0.886}^{\pm.004}$ | $0.347^{\pm.031}$ | $\mathbf{2.463}^{\pm.012}$ | $9.428^{\pm.085}$ |
| $\{0, 0, 0, 1\}$ | $0.580^{\pm.006}$ | $0.773^{\pm.005}$ | $0.860^{\pm.006}$ | $\underline{0.460}^{\pm.052}$ | $2.621^{\pm.013}$ | $9.033^{\pm.088}$ |
| $\{0.25, 0.25, 0.25, 0.25\}$ | $\underline{0.582}^{\pm.004}$ | $\underline{0.780}^{\pm.005}$ | $\underline{0.867}^{\pm.004}$ | $0.478^{\pm.052}$ | $\underline{2.614}^{\pm.016}$ | $\underline{9.201}^{\pm.092}$ |
| $\{0.1, 0.2, 0.3, 0.4\}$ **(Ours)** | $\mathbf{0.607}^{\pm.005}$ | $\mathbf{0.800}^{\pm.004}$ | $\mathbf{0.886}^{\pm.004}$ | $\mathbf{0.347}^{\pm.031}$ | $\mathbf{2.463}^{\pm.012}$ | $\mathbf{9.428}^{\pm.085}$ |

as "*waving*" are often omitted. Moreover, all three methods struggle with direction-related prompts (*e.g.*, "*forward*", "*right*") and left–right limb coordination, leading to motions that deviate from the specified details. In contrast, our method achieves higher-fidelity motion generation and consistently aligns with complex, fine-grained prompts.

### 4.3 ABLATION STUDY

**Effectiveness of Tri-domain Modeling.** As shown in the upper part of Tab. 3, we progressively add STM, HFA, and S-Fus on top of TME to reveal each module's contribution. When S-Fus is not used, tri-domain features are simply concatenated. Introducing STM or HFA alone already brings clear gains. STM improves R@1 by about 0.1 and reduces MM-Dist by enforcing realistic joint topology, while HFA yields the largest FID drop by modeling global trends and fine-grained dynamics via low- and high-frequency cues. Since spatial structure and frequency characteristics describe complementary aspects of motions, combining STM and HFA further enhances semantic alignment and fidelity. Building on these foundation, the full framework with S-Fus achieves the best performance, improving R@1 to 0.607, by adaptively weighting domain-specific cues rather than relying on simple concatenation.

**Impact of Frequency Components (Low vs. Low + High).** The lower part of Tab. 3 validates the effectiveness of the internal components of HFA. Removing the FFT branch results in a clear

Table 5: User study results.

| Method | Text-Motion Alignment | Overall Quality |
|---|---|---|
| MARDM (Meng et al., 2024) | $2.970^{\pm.108}$ | $3.268^{\pm.100}$ |
| MoMask (Guo et al., 2024) | $3.096^{\pm.112}$ | $3.289^{\pm.103}$ |
| SALAD (Hong et al., 2025) | $\underline{3.518}^{\pm.108}$ | $\underline{3.567}^{\pm.099}$ |
| Ours | $\mathbf{4.125}^{\pm.087}$ | $\mathbf{3.981}^{\pm.091}$ |

decline in both semantic alignment and motion fidelity, indicating that modeling the low-frequency components within a hybrid FFT–DWT frequency space enables the network to better capture global motion structures. Limiting HFA to the temporal domain (*"w/o joint"*) further degrades multiple metrics, demonstrating that joint-domain frequency modeling is crucial for capturing fine-grained spatial dynamics and ensuring coherent multi-joint coordination. Compared to TriC-Motion without HFA, using only the low-frequency band improves R@1 to 0.602, MM-Dist to 2.494, and reduces FID. Combining both frequency components achieves the best performance, optimizing coarse motion trends and fine-grained dynamics, further confirming HFA's capability for faithful and diverse motion generation.

**Ablation of Causal Intervention.** As shown in the upper part of Tab. 4, removing CCMD significantly degrades R-Precision and increases FID, demonstrating CCMD's role in guiding the model to reduce motion-irrelevant cues, thereby improving generation quality. Applying CCMD solely to the temporal domain yields measurable improvements in text–motion alignment, with R@1 enhanced to 0.580. Extending CCMD to both temporal and spatial domains further enhances text-motion alignment and reduces MM-Dist, achieving performance comparable to the *"post"* variant. TriC-Motion with full tri-domain configurations achieves the best performance, confirming that causal intervention is most effective when jointly applied across all three domains.

**Ablation on Layer-wise Loss Weights of $\mathcal{L}_{fcf}$.** We conduct an ablation on the layer-wise weights of $\mathcal{L}_{\text{fcf}}$, with results shown in the lower part of Tab. 4. The progressively increasing schedule $\{0.1, 0.2, 0.3, 0.4\}$ attains the best results. This demonstrates that assigning larger weights to deeper denoising blocks can suppress domain-specific confounders and provide causal guidance more effectively, resulting in a more stable denoising process and better overall performance.

## 4.4 USER STUDY

To validate the perceptual quality of motion generation, we conduct a user study following the protocol adopted by GMMotion (Tu et al.) and SALAD Hong et al. (2025). In this study, 38 participants compare our method against three representative baselines: MARDM (Meng et al., 2024), MoMask (Guo et al., 2024), and SALAD (Hong et al., 2025), which represent MAR paradigms, discrete AR models, and continuous diffusion frameworks, respectively. For each method, participants are presented with 15 video examples and evaluate them based on two criteria: Overall Quality and Text–Motion Alignment. All ratings are collected using a 5-point Likert scale ranging from 1 (poorest) to 5 (best). The results, summarized in Tab. 5, demonstrate that the motions generated by TriC-Motion are more preferred by humans in terms of overall visual quality and text–motion consistency compared to other methods.

## 5 CONCLUSION

We propose TriC-Motion, a novel diffusion-based framework for text-to-motion generation that integrates temporal, spatial, and frequency modeling with causal intervention. This framework features three modules, TME, STM, and HFA, for comprehensive domain-specific modeling. S-Fus then adaptively integrates these domain-specific cues, while TIJ injects semantic information into motion features. Furthermore, the framework incorporates CCMD to perform causal intervention during training, which suppresses motion-irrelevant cues caused by noise while enhancing real causal contributions. Extensive experiments demonstrate the state-of-the-art performance of TriC-Motion, particularly in R-Precision and MM-Dist, highlighting its powerful motion-generation capability.

## 6 ACKNOWLEDGEMENT

This work was supported by National Natural Science Foundation of China (No. 62376102) and Ant Group.

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

## A APPENDIX

### A.1 ETHICS STATEMENT

This work adheres to the ICLR Code of Ethics. In this study, no human subjects or animal experimentation were involved. All datasets used, including HumanML3D and SnapMoGen, were sourced in compliance with relevant usage guidelines, ensuring no violation of privacy. We have taken care to avoid any biases or discriminatory outcomes in our research process. No personally identifiable information was used, and no experiments were conducted that could raise privacy or security concerns. We are committed to maintaining transparency and integrity throughout the research process.

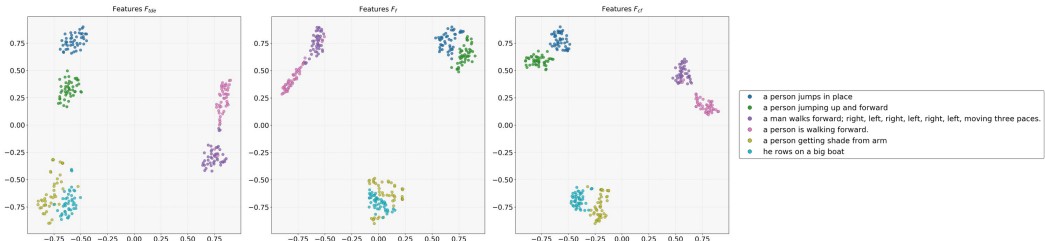

Figure A1: t-SNE Visualization of the the three feature types in TriC-Motion: motion-relevant features $F_f$, motion-irrelevant (confounding) features $F_{cf}$, and the final causally disentangled features $F_{tde}$ used for generation. The six text inputs are taken from the HumanML3D test set.

Table A1: FLOPs and average inference time (AIT) comparison across various methods.

| Method | FLOPs (G) | AIT (s) |
|---|---|---|
| MDM (Tevet et al., 2022) (DDIM-50step) | 325.25 | 1.5 |
| SALAD (Hong et al., 2025) | 233.83 | 3.0 |
| MARDM (Meng et al., 2024) | 23519.50 | 10.6 |
| MoMask (Guo et al., 2024) | 37.50 | 0.4 |
| Ours | 388.45 | 3.8 |

## A.2 REPRODUCIBILITY STATEMENT

We have made every effort to ensure that the results presented in this paper are reproducible. All the code will be released after the double-blind review process to ensure that the results can be reproduced and verified. The experimental setup, including training steps, model configurations, and hardware details, is described in detail in the paper. Additionally, the HumanML3D and SnapMoGen datasets are publicly available, ensuring consistent and reproducible evaluation results. We believe these measures will enable other researchers to reproduce our work and further advance the field.

## A.3 LLMS USAGE STATEMENT

This paper utilizes Large Language Models (LLMs) solely as a tool for polishing the texts, thereby refining the paper and improving readability. All ideas, research concepts, research methodology, experiments, figures, and visualizations are conducted without the use of LLMs.

## A.4 T-SNE VISUALIZATION OF CCMD

As shown in Fig. A1, we present t-SNE visualizations of six different text inputs, from the HumanML3D test dataset, processed by TriC-Motion, showing the motion-relevant features $F_f$, motion-irrelevant features $F_{cf}$, and the final features contributing to generation $F_{tde}$ obtained through causal modeling. As shown in the t-SNE visualizations on our website, for certain complex or easily confusable motion texts, their features $F_f$ overlap in the space. After being modeled by the CCMD module based on causal learning, the disentangled features $F_{tde}$ clearly demonstrate that the features corresponding to each text become separated, and features within the same category are more clustered. This indicates that after removing the confounding information and motion-relevant factors, the model generates more accurate and less ambiguous motion features based on the text, thereby improving the consistency between generated motion and text as well as the fidelity of the motion.

## A.5 COMPARISON OF COMPUTATIONAL COST, INFERENCE TIME AND MODEL PARAMETERS

As showin in Tab. A1, we present the FLOPs and average inference time (AIT) measured over 100 samples on a single NVIDIA RTX 3090 GPU for several representative methods. Parameter sizes are summarized in Tab. A2. TriC-Motion requires 388.45 GFLOPs, which is only slightly

Table A2: Parameter size comparison across various methods.

| Method | Params (M) |
|---|---|
| MDM (Tevet et al., 2022) | 17.88 |
| MLD (Chen et al., 2023b) | 26.38 |
| ReMoDiffuse (Zhang et al., 2023b) | 46.97 |
| MoMask (Guo et al., 2024) | 44.85 |
| SALAD (Hong et al., 2025) | 10.10 |
| MARDM (Meng et al., 2024) | 310.09 |
| Ours | 13.86 |

Table A3: Sensitivity analysis of the perceptual loss $\mathcal{L}_p$ (upper part) and causal loss $\mathcal{L}_{fcf}$ (lower part). Here, $\alpha$ and $\beta$ denote the weights of $\mathcal{L}_p$ and $\mathcal{L}_{fcf}$, respectively. The default setting (Ours) corresponds to $\alpha = 1.0, \beta = 10$.

| $\alpha$ | $\beta$ | R-Precision ↑ | | | FID ↓ | MM-Dist ↓ | Diversity → |
|---|---|---|---|---|---|---|---|
| | | Top 1 | Top 2 | Top 3 | | | |
| 1.0 | 10 | $0.607^{\pm.005}$ | $0.800^{\pm.004}$ | $\mathbf{0.886}^{\pm.004}$ | $\mathbf{0.347}^{\pm.031}$ | $\mathbf{2.463}^{\pm.012}$ | $9.428^{\pm.085}$ |
| 1.0 | 1 | $0.597^{\pm.006}$ | $0.787^{\pm.006}$ | $0.867^{\pm.005}$ | $0.416^{\pm.035}$ | $2.571^{\pm.023}$ | $9.917^{\pm.066}$ |
| 1.0 | 20 | $0.600^{\pm.006}$ | $0.794^{\pm.005}$ | $0.878^{\pm.004}$ | $0.425^{\pm.037}$ | $2.517^{\pm.017}$ | $9.229^{\pm.074}$ |
| 0.5 | 10 | $\mathbf{0.609}^{\pm.005}$ | $\mathbf{0.803}^{\pm.004}$ | $0.882^{\pm.004}$ | $0.349^{\pm.032}$ | $2.485^{\pm.017}$ | $\mathbf{9.464}^{\pm.071}$ |
| 2 | 10 | $0.599^{\pm.008}$ | $0.793^{\pm.005}$ | $0.881^{\pm.004}$ | $0.418^{\pm.035}$ | $2.527^{\pm.018}$ | $9.188^{\pm.107}$ |

higher than our baseline (MDM), and achieves an AIT of 3.8 s, remaining comparable to latent-space diffusion models such as SALAD (3.0 s) while being considerably faster than MARDM (10.6 s). The model is also highly compact, containing only 13.86M parameters, which is significantly smaller than many strong baselines including MoMask (44.85M), ReMoDiffuse (46.97M), MLD (26.38M), and MARDM (310.09M).

The additional computation primarily comes from explicit tri-domain modeling, which cannot be replicated by simply scaling a single-domain network. Despite incorporating temporal, spatial, and frequency branches, the overall architecture remains deliberately lightweight. Furthermore, TriC-Motion conducts diffusion directly in the raw motion space, where multi-step denoising inherently incurs higher cost. This is a well-known limitation shared across diffusion-based motion generators. Nevertheless, the proposed tri-domain design yields clear and consistent benefits, improving R@1 to 0.612 and enhancing motion fidelity on both HumanML3D and SnapMoGen.

It is also worth noting that the causal module is used only during training and introduces no inference-time overhead, and all tri-domain branches are efficient and parallelizable.

Prior works (StableMoFusion (Huang et al., 2024a) and Light-T2M (Zeng et al., 2025)) demonstrate that advanced samplers (*e.g.*, DPM-Solver (Lu et al., 2022), UniPC (Zhao et al., 2023)) can reduce diffusion to as few as 10 steps, yielding substantial acceleration with minimal quality degradation. These findings indicate that TriC-Motion can be efficiently accelerated using modern sampling strategies, and in future work we will further explore such techniques while also investigating diffusion in a more compact latent space to optimize the inference process.

## A.6 SENSITIVITY ANALYSIS OF LOSS WEIGHTING

We perform a comprehensive sensitivity analysis on both the perceptual loss and the causal loss, and the results are presented in Tab. A3. The findings consistently show that TriC-Motion remains stable under a wide range of loss-weight configurations. When the weight of the perceptual loss varies from 1 to 20, the changes in R-Precision remain minor, with values of 0.597 and 0.607, and the corresponding FID values vary only slightly from 0.347 to 0.425. No sign of instability or significant performance degradation is observed in this process. A similar phenomenon appears when adjusting the weight of the causal loss from 0.5 to 2, where the resulting performance remains

Table A4: The performance comparison on HumanML3D dataset between our TriC-Motion without $\mathcal{L}_p$ and the previous state-of-the-art method, SALAD (Hong et al., 2025).

| Method | R@1 ↑ | R@2 ↑ | R@3 ↑ | MM-Dist ↓ |
|---|---|---|---|---|
| GT | $0.511^{\pm.003}$ | $0.703^{\pm.003}$ | $0.797^{\pm.002}$ | $2.974^{\pm.008}$ |
| SALAD (Hong et al., 2025) | $0.581^{\pm.003}$ | $0.769^{\pm.003}$ | $0.857^{\pm.002}$ | $2.649^{\pm.009}$ |
| Ours (w/o $\mathcal{L}_p$) | $\underline{0.585}^{\pm.005}$ | $\underline{0.772}^{\pm.006}$ | $\underline{0.863}^{\pm.004}$ | $\underline{2.553}^{\pm.019}$ |
| Ours | $\mathbf{0.607}^{\pm.005}$ | $\mathbf{0.800}^{\pm.004}$ | $\mathbf{0.886}^{\pm.004}$ | $\mathbf{2.463}^{\pm.012}$ |

Table A5: Performance comparison on HumanML3D dataset under the CLaM evaluator.

| Methods | R@1 ↑ | R@2 ↑ | R@3 ↑ |
|---|---|---|---|
| GT | $0.738^{\pm.200}$ | $0.870^{\pm.200}$ | $0.917^{\pm.100}$ |
| MLD (Chen et al., 2023b) | $0.599^{\pm.003}$ | $0.760^{\pm.002}$ | $0.831^{\pm.002}$ |
| MotionDiffuse (Zhang et al., 2024a) | $0.645^{\pm.004}$ | $0.803^{\pm.003}$ | $0.868^{\pm.003}$ |
| T2M-GPT (Zhang et al., 2023a) | $0.676^{\pm.003}$ | $0.820^{\pm.004}$ | $0.878^{\pm.004}$ |
| MotionGPT (Jiang et al., 2023) | $0.478^{\pm.002}$ | $0.655^{\pm.002}$ | $0.752^{\pm.002}$ |
| T2M (Guo et al., 2022) | $0.577^{\pm.003}$ | $0.730^{\pm.002}$ | $0.804^{\pm.002}$ |
| MoMask (Guo et al., 2024) | $0.715^{\pm.002}$ | $0.856^{\pm.002}$ | $0.909^{\pm.001}$ |
| SALAD (Guo et al., 2024) | $\underline{0.776}^{\pm.003}$ | $\underline{0.902}^{\pm.002}$ | $\underline{0.943}^{\pm.001}$ |
| **Ours** | $\mathbf{0.786}^{\pm.006}$ | $\mathbf{0.912}^{\pm.005}$ | $\mathbf{0.952}^{\pm.003}$ |

highly consistent. These observations indicate that the causal intervention branch does not introduce additional sensitivity and that the overall optimization is robust. The results therefore confirm that TriC-Motion is not sensitive to the weighting of these loss terms.

## A.7 ABLATION STUDY OF PERCEPTUAL LOSS $\mathcal{L}_p$ AND CROSS-EVALUATOR VALIDATION

We further analyze the role of the perceptual loss $\mathcal{L}_p$ to ensure that it does not induce unintended coupling with the HumanML3D evaluator. The use of a perceptual objective in a learned feature space follows a well-established practice in generative modeling, where such objectives accelerate convergence and improve perceptual quality, particularly in diffusion-based image and video generation. Although incorporating this perceptual signal indeed brings a moderate improvement, its presence is not the source of our R-Precision gains. As shown in Tab. A4, removing the perceptual loss still yields competitive results compared to strong baselines (*e.g.*, SALAD (Hong et al., 2025)), including solid R-Precision and MM-Dist performance. This confirms that the central improvements originate from our tri-domain modeling and causal intervention design, rather than from coupling between the loss and the HumanML3D evaluator.

We additionally evaluate the same trained model using CLaM (Chen et al., 2024b), a more advanced text–motion evaluator whose embedding space is entirely different from the one used during training. This setup eliminates any possible overlap between the perceptual loss feature space and the evaluation space. As shown in Tab. A5, despite the shift to this substantially stronger evaluator, our method consistently achieves superior R-Precision, surpassing both the ground truth and strong baselines such as SALAD (Hong et al., 2025) by a large margin. Even under the condition where different evaluators are used during training and inference, these performance improvements persist. This demonstrates that our performance gain does not rely on the alignment between the perceptual loss space and the evaluation feature space, nor does it stem solely from the perceptual loss term. Instead, it originates from the combined innovative design of tri-domain modeling, causal learning, and other contributions.

## A.8 SUPPLEMENTARY VISUAL MATERIAL

To facilitate comprehensive evaluation, we provide a dedicated website hosting all supplementary qualitative materials, including motion visualizations, side-by-side comparisons with state-of-the-

art methods, and t-SNE visualization: https://sites.google.com/view/tric-motion-iclr2026. These materials complement the main paper by offering additional perspectives on motion fidelity and semantic alignment. Importantly, the website strictly adhere to the double-blind review principles of the ICLR 2026 conference, ensuring the anonymity of the website.

