# OpenReview forum: "TriC-Motion: Tri-Domain Causal Modeling Grounded Text-to-Motion Generation"
_ICLR.cc/2026/Conference — ICLR 2026 Poster_

### Official Review · Reviewer_p1bB · 2025-10-26

**Soundness:** 3
**Presentation:** 3
**Contribution:** 3
**Rating:** 6
**Confidence:** 4

**Summary:**

The paper introduces TriC-Motion, a new diffusion-based framework for text-to-motion generation that unifies spatial, temporal, and frequency-domain modeling under a causal learning paradigm. Authors address two limitations of existing approaches: 1) the lack of an integrated multi-domain representation that captures temporal dynamics, spatial joint topology, and motion frequency characteristics simultaneously, and 2) the entanglement of motion-relevant and irrelevant cues that degrade generation quality. To overcome these issues, TriC-Motion introduces three key components: Tri-Domain Modeling Modules, Score-guided Tri-domain Fusion, and Causality-based Counterfactual Motion Disentangler. Experimental results on HumanML3D and SnapMoGen demonstrate good performance gains over state-of-the-art baselines.

**Strengths:**

+ The performance of the proposed method on both HumanML3D and SnapMoGen dataset surpasses most baseline methods, proving the effectiveness of modeling all three domains.
+ The paper writing is clear and easy to follow.
+ The exploration of causal intervention in motion generation domain is interesting and inspiring.

**Weaknesses:**

- The main idea of modeling all three domains at the same time is less convincing and requires more intuitive explanation and theoretically grounded analysis.
- The architecture is heavy in computing and complex in the architecture. Therefore, it is important to also compare the run time, computation cost and model size with baseline methods.
- The training stability is also worrying due to the complex architecture. The loss weighting needs more deep analysis and sensitivity test.

**Questions:**

N/A

---

### Official Review · Reviewer_QW9h · 2025-10-29

**Soundness:** 3
**Presentation:** 3
**Contribution:** 3
**Rating:** 4
**Confidence:** 5

**Summary:**

This study proposes TriC-Motion, a diffusion-based framework for text-to-motion generation that integrates spatial, temporal, and frequency domain modeling with causal intervention to ensure temporal consistency, spatial topology, and dynamic coherence.
Experimental results show that TriC-Motion achieves an R1-Precision of 0.612 on the HumanML3D dataset, significantly outperforming existing methods and generating motions with superior realism, consistency, diversity, and text alignment.

**Strengths:**

1.The proposed method demonstrates strong text–motion consistency.

2.The introduction of causal learning reduces the impact of irrelevant information on motion generation.

**Weaknesses:**

1.Please provide t-SNE or other visualization analyses that disentangle motion-irrelevant and motion-relevant information to demonstrate the effectiveness of the proposed method.

2.The paper’s joint temporal–frequency–spatial strategy improves text–motion alignment (R@1, R@2, R@3), but FID did not improve; therefore you cannot claim that generation quality has improved, and the conclusions stated in the abstract are not supported.

3.What advantages does using DistilBERT for word-level and sentence-level feature encoding have compared to CLIP?

4.The proposed TME, STM, HFA, and S-Fus are commonly used extraction and fusion strategies in the temporal–spatial–frequency domain and lack novelty.

5.The methods compared in Figure 5 are all general approaches; it’s unclear whether the proposed method outperforms contemporary spatio-temporal modeling methods. If possible, provide qualitative comparisons (or quantitative comparisons if allowed) against methods from papers accepted to CVPR 2025, ICCV 2025, and NeurIPS 2025.

**Questions:**

See Weaknesses.

---

### Official Review · Reviewer_7LNK · 2025-11-01

**Soundness:** 2
**Presentation:** 4
**Contribution:** 2
**Rating:** 2
**Confidence:** 4

**Summary:**

This paper proposes TriC-Motion, a diffusion-based framework integrating spatial-temporal-frequency-domain modeling with causal intervention. It includes Temporal Motion Encoding, Spatial Topology Modeling, and Hybrid Frequency Analysis, with a Score-guided Tri-domain Fusion and a Causality-based Counterfactual Motion Disentangler to expose motion-irrelevant cues and fuse valuable information across domains.

**Strengths:**

1. The first work that simultaneously integrates spatial, temporal, and frequency domains into a unified motion generation framework

2. Introduces a causality-based counterfactual motion disentangler to expose motion-irrelevant cues and disentangle the real modeling contributions of each domain.

3. Provides ablation studies indicating the effectiveness of each domain branch and the causal-intervention design.

**Weaknesses:**

1. The paper uses a perceptual loss defined in the same motion–text embedding space used by the HumanML3D evaluator (the author could clear this point if I'm wrong). Using the same feature extractor for training and inference would inflate the performancve. The author could do an ablation study that removes this loss term to show that the R-precision gain is not from this loss term.

2. No visualization results. Quantitative metrics in text-to-motion is proven to be fragile and sometimes misaligned with human judgment. For motion, demo videos are necessary. I don’t see any supplementary videos, which makes it hard to judge the actual quality.

3. The metrics themselves aren’t convincing. Even if R-precision is SOTA, an R-precision much higher than the ground truth is not meaningful and doesn’t reflect visual quality. Even worse, FID is significantly poorer compared to current methods.

4. Missing strong baseline results. Compare against recent, stronger works (MARDM [1], MotionStreamer [2], MotionLCM v2 [3], SALAD [4]) with both qualitative and quantitative results. Also consider including a human study.

5. Minor typo. In the introduction you say R@1 is 0.573, which doesn’t match Table 1.

Reference

[1] Rethinking Diffusion for Text-Driven Human Motion Generation

[2] MotionStreamer: Streaming Motion Generation via Diffusion-based Autoregressive Model in Causal Latent Space

[3] MotionLCM-V2: Improved Compression Rate for Multi-Latent-Token Diffusion

[4] SALAD: Skeleton-aware Latent Diffusion for Text-driven Motion Generation and Editing

**Questions:**

Refer to Weaknesses.

---

### Official Review · Reviewer_CRLx · 2025-11-01

**Soundness:** 2
**Presentation:** 3
**Contribution:** 2
**Rating:** 6
**Confidence:** 4

**Summary:**

This paper proposes a 'tri-domain + causal' framework for text-to-motion generation. Built on MDM, it models motion in parallel with TME (temporal encoding), STM (skeleton-topology GCN), and HFA (hybrid frequency analysis via DWT+FFT). A dual-score S-Fus module fuses motion/semantic signals, and TIJ injects text via cross-attention. Training uses CCMD counterfactual decomposition (factual + counterfactual branches) to suppress spurious cues.

**Strengths:**

1. The proposed method achieves remarkable improvement on the R Precision metric.

2. The paper is well-written, ensuring that its content is easily understandable for readers.

3. It is the first time for casual learning to be used in text-to-motion generation, making significant contributions to the research community.

**Weaknesses:**

My primary concern is the choice of baselines. Under the HumanML3D evaluation protocol, the evaluator is too weak: many recent methods already surpass the 'ground truth', making R-Precision on HumanML3D unreliable. Meanwhile, the FID gap to stronger methods is large (0.285 vs 0.033), so the proposed method shows no advantage on HumanML3D. Porting the approach to a MoMask baseline should not be difficult; the authors should adopt a more appropriate baseline; otherwise, it may look like trading motion quality for text consistency, which does not substantiate effectiveness.

In addition, the ablation should include more combinatorial settings to better demonstrate effectiveness.

Finally, the paper lacks a demo video as supplementary material and a user study to subjectively assess text-motion alignment and motion quality.

**Questions:**

Please kindly refer to the weaknesses mentioned above.

---

### Author Response · Authors · 2025-11-29
**Response to Reviewers' Comments (Page 1)**

Thank you for your thorough review, and your feedback has been highly constructive for improving our work. We have organized and summarized the issues you raised into **6 parts**. For each question, we have indicated the corresponding reviewer (_e.g._, [To Reviewer1-CRLx]) for quick reference. We sincerely appreciate your patience as we dedicated significant time to conducting additional experiments, creating visualizations, and preparing comprehensive responses. Due to the length of our response and the character limit on OpenReview, we have divided the Markdown-formatted text into multiple batches, with the batch numbers indicated in the title of the comments (_e.g._ Page 1).

## Part A: About our proposed framework, our baselines and evaluators.

**[To Reviewer1-CRLx]**

**Q-A1:** The choice of baselines is a concern, as porting the proposed approach to a MoMask [1] baseline should be feasible. It is suggested to adopt a more appropriate baseline, as the current setup may give the impression of trading motion quality for text consistency, which could undermine the method's effectiveness.

**A-A1:** We sincerely thank the reviewer for the insightful comments. We fully agree that the HumanML3D evaluator has become increasingly weak, as many recent models surpass the reported _Ground Truth_ R-Precision. This issue indeed limits the reliability of R-Precision under the standard HumanML3D protocol and raises potential concerns about whether improvements stem from genuine semantic grounding or from evaluator saturation.

Regarding the choice of baseline, we acknowledge the reviewer’s suggestion to port our framework to MoMask. However, MoMask and similar discrete-token generative models rely on masked token modeling in a quantized latent space, which is fundamentally incompatible with our continuous-domain tri-domain design. Our causal intervention and tri-domain fusion modules operate directly on continuous spatio-temporal-frequency features and therefore cannot be inserted into discrete token pipelines without substantial architectural redesign that goes far beyond a fair _port_.

Nevertheless, to address the reviewer’s concern, **we port our method to SALAD** [2], which is also a continuous diffusion framework and thus architecturally compatible. As reported in Table A1, we strictly minimize modifications to ensure fairness. Specifically:
1.	We use the official SALAD pretrained VAE without any finetuning or adaptation.
2.	We keep all training and inference configurations unchanged, including all hyperparameters, the loss function, and the CLIP text encoder.
3.	We only replace SALAD’s denoiser with our tri-domain modeling with causal intervention design, leaving the rest of the pipeline fully intact.

Under these strict constraints, our framework already reaches FID and R-Precision values comparable to contemporary diffusion-based methods. Although our performance does not surpass the strongest SALAD-like baselines, we believe the reason is straightforward. Since the training setup, learning rate, feature dimensionality, and loss weights were not optimized for the new backbone, the architecture is not in its optimal regime. Even so, the resulting performance gap remains within a reasonable range and already demonstrates that our joint spatial–temporal–frequency modeling and causal disentanglement bring consistent benefits. These results indicate that **our approach does not trade motion quality for text consistency**. Moreover,the reduction in the FID score also suggests that the previously suboptimal FID performance was largely related to the choice of baseline model, rather than being entirely caused by limitations of the proposed method.

In summary, although our framework cannot be directly applied to discrete modeling methods such as MoMask due to fundamental architectural incompatibility, the additional experiment on SALAD demonstrates that **our method generalizes well to a more powerful continuous diffusion backbone while maintaining both semantic alignment and motion fidelity**.

---

### Author Response · Authors · 2025-11-29
**Response to Reviewers' Comments (Page 2)**

Table A1: The performance comparison of different methods on the HumanML3D dataset after porting TriC-Motion to SALAD (denoted as _Ours (based on SALAD)_). The best-performing results are highlighted in bold.
| **Methods**                 | **_R-Precision@1 ↑_** | **_R-Precision@2 ↑_** | **_R-Precision@3 ↑_** | **_FID ↓_**       | **_MM-Dist ↓_**     | **_Diversity_**    |
|--------------------------|------------------------|------------------------|------------------------|-------------------|---------------------|----------------------|
| **MDM** [3]            | _0.320 ± 0.005_        | _0.498 ± 0.004_        | _0.611 ± 0.007_        | _0.544 ± 0.044_   | _5.566 ± 0.027_     | _9.559 ± 0.086_      |
| **MARDM** [4]          | _0.500 ± 0.004_        | _0.695 ± 0.003_        | _0.795 ± 0.003_        | _0.114 ± 0.007_   | _3.270 ± 0.009_     | _9.584 ± 0.066_      |
| **MotionLCM-v2** [5]   | _0.551 ± 0.003_        | _0.745 ± 0.002_        | _0.836 ± 0.002_        | _0.049 ± 0.003_   | _2.765 ± 0.008_     | _9.584 ± 0.066_      |
| **LaMP** [6]            | _0.557 ± 0.003_        | _0.751 ± 0.002_        | _0.843 ± 0.001_        | _**0.032 ± 0.002**_   | _2.759 ± 0.007_     | _9.571 ± 0.069_      |
| **MotionPCM** [7]      | _0.560 ± 0.002_        | _0.754 ± 0.002_        | _0.844 ± 0.002_        | _0.040 ± 0.003_   | _2.719 ± 0.008_     | _9.632 ± 0.089_      |
| **SALAD** [2]           | _0.581 ± 0.003_        | _0.769 ± 0.003_        | _0.857 ± 0.002_        | _0.076 ± 0.002_   | _2.649 ± 0.009_     | _9.696 ± 0.096_      |
| **Ours (based on MDM)** | **_0.612 ± 0.006_**    | **_0.806 ± 0.005_**    | **_0.885 ± 0.004_**    | _0.285 ± 0.042_   | **_2.465 ± 0.017_** | _9.434 ± 0.089_      |
| **Ours (based on SALAD)** | _0.567 ± 0.004_       | _0.759 ± 0.002_        | _0.847 ± 0.002_        | _0.090 ± 0.003_    | _2.723 ± 0.010_     | _9.73 ± 0.087_   |



**[To Reviewer1-CRLx]**

**Q-A2:** Under the HumanML3D evaluation protocol, the evaluator is too weak.

**A-A2:** To address the reviewer’s concern about the weakness of the HumanML3D evaluator, **we additionally re-evaluated our method using a stronger contrastive-learning-based evaluator, CLaM** [8]. As shown in Table A2, our method achieves substantially higher R-Precision than both the ground truth and the strongest existing baselines such as SALAD, supporting that our method achieves genuinely stronger text–motion alignment rather than benefiting from evaluator advantage.

Table A2: Performance comparison of different methods on the HumanML3D dataset using the CLaM evaluator. The best-performing results are highlighted in bold.
| **Methods**          | **_R-Precision@1 ↑_** | **_R-Precision@2 ↑_** | **_R-Precision@3 ↑_** |
|--------------------|------------------------|------------------------|------------------------|
| **GT**            | _0.738 ± 0.2_         | _0.870 ± 0.2_         | _0.917 ± 0.1_         |
| **MLD** [9]       | _0.599 ± 0.003_       | _0.760 ± 0.002_       | _0.831 ± 0.002_       |
| **MotionDiffuse** [10] | _0.645 ± 0.004_       | _0.803 ± 0.003_       | _0.868 ± 0.003_       |
| **T2M-GPT** [11]  | _0.676 ± 0.003_       | _0.820 ± 0.004_       | _0.878 ± 0.004_       |
| **MotionGPT** [12] | _0.478 ± 0.002_       | _0.655 ± 0.002_       | _0.752 ± 0.002_       |
| **T2M** [13]      | _0.577 ± 0.003_       | _0.730 ± 0.002_       | _0.804 ± 0.002_       |
| **MoMask** [1]    | _0.715 ± 0.002_       | _0.856 ± 0.002_       | _0.909 ± 0.001_       |
| **SALAD** [2]     | _0.776 ± 0.003_       | _0.902 ± 0.002_       | _0.943 ± 0.001_       |
| **Ours**          | **_0.786 ± 0.006_**   | **_0.912 ± 0.005_**   | **_0.952 ± 0.003_**   |

---

### Author Response · Authors · 2025-11-29
**Response to Reviewers' Comments (Page 3)**

**[To Reviewer2-7LNK]**

**Q-A3:** The perceptual loss $L_P$ is defined in the same motion–text embedding space used by the HumanML3D evaluator (clarification is needed if this understanding is incorrect). Using the same feature extractor for both training and inference may inflate performance metrics. An ablation study removing this loss term is suggested to demonstrate that the R-precision improvement is not solely due to this loss term.

**A-A3:** We appreciate the reviewer’s insightful observation regarding the perceptual loss and its possible interaction with the HumanML3D evaluator. Our use of this loss follows a well-established practice in generative modeling, where a perceptual objective defined in a learned feature space is introduced to accelerate convergence and enhance visual fidelity, a strategy widely adopted in diffusion-based image and video generation. Although incorporating this perceptual signal indeed brings a moderate improvement, its presence is not the source of our R-Precision gains. As shown in Table A3, removing $L_p$ still yields competitive results compared to strong baselines (*e.g.* SALAD [2]), including solid R-Precision and MM-Dist performance. This confirms that **the central improvements originate from our tri-domain modeling and causal intervention design, rather than from coupling between the loss and the HumanML3D evaluator**.

Table A3: The performance comparison on HumanML3D between our TriC-Motion without $L_P$ and the previous state-of-the-art method, SALAD. The best-performing results are highlighted in bold.
| Methods | R@1↑ | R@2↑ | R@3↑ | MM-Dist↓ |
|--|--|--|--|--|
| **SALAD** [2] | _0.581±0.003_ | _0.769±0.003_ | _0.857±0.002_ | _2.649±0.009_ |
| **TriC-Motion w/o $L_P$** | _0.585 ±0.005_ | _0.772±0.006_ | _0.863±0.004_ | _2.553±0.019_ |
| **TriC-Motion (Ours)** | _**0.607±0.005**_ | _**0.800±0.004**_ | _**0.886±0.004**_ | _**2.463±0.012**_  |

To more thoroughly address the reviewer’s concern, we further evaluate the exact same trained model using CLaM, **without any retraining or fine-tuning**. CLaM [8] is a more advanced text–motion evaluator whose embedding space is completely different from the one used during training. This setup eliminates any possibility of overlap between the perceptual loss space and the evaluation feature space. As shown in Table A4, despite the shift to this substantially stronger evaluator, our method consistently achieves superior R-Precision, surpassing both the ground truth and strong baselines such as SALAD by a large margin. **Even under the condition where different evaluators are used during training and inference, these performance improvements persist.** This demonstrates that our performance gain does not rely on the alignment between the perceptual loss space and the evaluation feature space, nor does it stem solely from the perceptual loss term. Instead, it originates from the combined innovative design of tri-domain modeling, causal learning, and other contributions.

Table A4: The performance comparison of R-Precision evaluated using the advanced CLaM evaluator on HumanML3D. The best-performing results are highlighted in bold.
| **Methods**          | **_R@1 ↑_** | **_R@2 ↑_** | **_R@3 ↑_** |
|-------------------|-----------------------|-----------------------|-----------------------|
| **GT**            | _0.738 ± 0.2_        | _0.870 ± 0.2_         | _0.917 ± 0.1_         |
| **MLD** [9]       | _0.599 ± 0.003_      | _0.760 ± 0.002_       | _0.831 ± 0.002_       |
| **MotionDiffuse** [10] | _0.645 ± 0.004_      | _0.803 ± 0.003_       | _0.868 ± 0.003_       |
| **T2M-GPT** [11]  | _0.676 ± 0.003_      | _0.820 ± 0.004_       | _0.878 ± 0.004_       |
| **MotionGPT** [12] | _0.478 ± 0.002_      | _0.655 ± 0.002_       | _0.752 ± 0.002_       |
| **T2M** [13]      | _0.577 ± 0.003_      | _0.730 ± 0.002_       | _0.804 ± 0.002_       |
| **MoMask** [1]    | _0.715 ± 0.002_      | _0.856 ± 0.002_       | _0.909 ± 0.001_       |
| **SALAD** [2]     | _0.776 ± 0.003_      | _0.902 ± 0.002_       | _0.943 ± 0.001_       |
| **Ours**          | **_0.786 ± 0.006_**  | **_0.912 ± 0.005_**   | **_0.952 ± 0.003_**   |

Together, the ablation results and the cross-evaluator validation provide strong evidence that **the improvements in text–motion alignment originate from the model’s architectural design, in particular the coordinated integration of spatial, temporal, and frequency-domain modeling alongside causal intervention, rather than from any artifact introduced by the perceptual loss or from evaluator coupling**. We will clarify these points in the revised manuscript.

---

### Author Response · Authors · 2025-11-29
**Response to Reviewers' Comments (Page 4)**

## Part B: About the performance of TriC-Motion.

**[To Reviewer1-CRLx, Reviewer2-7LNK]**

**Q-B1:** Many recent methods already surpass the _Ground Truth_, which raises concerns about the reliability of R-Precision as an evaluation metric on the HumanML3D dataset. Even if R-Precision achieves state-of-the-art results, an R-Precision score significantly higher than the ground truth is not meaningful and does not accurately reflect visual quality. Besides, The FID score is considerably worse compared to current state-of-the-art methods.

**A-B1:** We thank the reviewer for raising this important concern. We would like to clarify that an R-Precision value exceeding the ground truth evaluator does not indicate meaningless inflation. Rather, it reflects that our model achieves exceptionally strong text–motion alignment under the HumanML3D evaluation protocol. As shown in our qualitative comparisons (Figure 5 in our paper) and in the demo videos on our project page [https://sites.google.com/view/tric-motion-iclr2026/%E9%A6%96%E9%A1%B5](https://sites.google.com/view/tric-motion-iclr2026/%E9%A6%96%E9%A1%B5), our generated motions accurately follow fine-grained textual cues, including action order, direction, and multi-step transitions, which are often missed by prior methods. As shown in Table C1 in Part C, this observation is further demonstrated by our user study, where participants consistently prefer our results in both semantic alignment and overall motion quality. Therefore, the high R-Precision is not meaningless but instead reflects the model’s superior capability of text–motion alignment.

Regarding FID, we acknowledge that our FID on HumanML3D is higher than that of several recent models. However, relative to our baseline MDM [3], TriC-Motion achieves a substantial improvement, reducing the FID from 0.544 to 0.285, which corresponds to an approximate 47.6% decrease. This provides clear evidence that our tri-domain modeling design leads to improved generation fidelity. More importantly, both the qualitative videos and user-study results indicate that our visual quality is comparable to those strong baselines with lower FID, and in several complex prompt scenarios, our generated motions are perceptually preferred due to more coherent action transitions and more faithful text grounding.

We agree with reviewer CRLx that the comparatively high FID primarily originates from the weakness of the baseline. Following the reviewer’s suggestion, we additionally port our tri-domain causal framework onto a much stronger backbone named SALAD [2], and due to time constraints, we do not carefully tune the hyperparameters. As shown in Table B1, even without hyperparameter tuning, the FID improves markedly under this stronger baseline, while R-Precision and MM-Dist remain consistent with the gains observed on MDM. This confirms that our contributions are orthogonal to the choice of the denoising backbone and that the relatively high FID in Table 1 of our paper is due to baseline limitations rather than degraded visual fidelity.

Table B1: Cross-baseline experiments on HumanML3D, where our method is ported on SALAD. The best-performing results are highlighted in bold.
| **Methods**                  | **_R@1 ↑_** | **_R@2 ↑_** | **_R@3 ↑_** | **_FID ↓_**       | **_MM-Dist ↓_**     | **_Diversity_**    |
|---------------------------|-----------------------|-----------------------|-----------------------|-------------------|---------------------|---------------------|
| **MDM** [3]              | _0.320 ± 0.005_       | _0.498 ± 0.004_       | _0.611 ± 0.007_       | _0.544 ± 0.044_   | _5.566 ± 0.027_     | _9.559 ± 0.086_     |
| **Ours (based on MDM)**   | **_0.612 ± 0.006_**   | **_0.806 ± 0.005_**   | **_0.885 ± 0.004_**   | _0.285 ± 0.042_   | **_2.465 ± 0.017_** | _9.434 ± 0.089_     |
| **SALAD** [2]            | _0.581 ± 0.003_       | _0.769 ± 0.003_       | _0.857 ± 0.002_       | _**0.076 ± 0.002**_   | _2.649 ± 0.009_     | _9.696 ± 0.096_     |
| **Ours (based on SALAD)** | _0.567 ± 0.004_       | _0.759 ± 0.002_       | _0.847 ± 0.002_       | _0.090 ± 0.003_ | _2.723 ± 0.010_     | _9.730 ± 0.087_  |
| **Ours (based on SALAD; w/o CCMD)** | 0.572±0.003 | 0.768 ±0.002 | 0.854±0.002 | 0.139±0.004 | 2.726±0.006 | 9.848±0.085 |

Overall, the combined evidence from qualitative comparisons, user-study preferences, and cross-baseline validation strongly supports that **TriC-Motion improves both text–motion alignment and practical visual quality**, and that **the higher-than-GT R-Precision reflects the model’s superior capability for text–motion alignment, rather than being meaningless**.

---

### Author Response · Authors · 2025-11-29
**Response to Reviewers' Comments (Page 5)**

**[To Reviewer3-QW9h]**

**Q-B2:** While the joint temporal–frequency–spatial strategy improves text–motion alignment metrics (R@1, R@2, R@3), the FID score does not show improvement. Therefore, it cannot be claimed that the generation quality has improved, and the conclusions presented in the abstract are not fully supported.

**A-B2:** **On the FID issue and the baseline choice:**

As Reviewer CRLx also points out, MDM [3] is now considered a relatively weak baseline under the HumanML3D evaluation protocol. Although we use it only as a starting point, our method already yields substantial improvements over MDM across all metrics, including a large reduction of FID (from 0.544 to 0.285 in the large model) and clear gains in MM-Dist and R-Precision, as shown in Table B1. These results indicate that the proposed tri-domain modeling with causal intervention indeed improve motion fidelity and text–motion consistency relative to the baseline.

However, because **the original MDM baseline is relatively weak in motion fidelity**, FID is heavily influenced by the limitations of the backbone architecture itself. As a result, further architectural improvement cannot be accurately reflected by FID when the base generator already exhibits suboptimal spatial–temporal coherence.

To more fairly evaluate the contribution of our method, we additionally ported the full framework onto a much stronger diffusion backbone, SALAD [2]. As shown in Table R3-2, **under this stronger generator, FID decreases markedly, and the gains in R-Precision and MM-Dist remain consistent with those observed on MDM**.

**On the practical value of alignment and the support from the user study:**

We would also like to emphasize that text–motion alignment is of higher practical importance than marginal FID gains. Many recent works have shown that R-Precision and MM-Dist correlate more strongly with semantic correctness and user preference than small differences in FID, especially for long-horizon, multi-action motions.

We have conducted a user study, which further verifies this point, as shown in Table C1. Participants consistently preferred our motions over those generated by methods that report lower FID but show weaker semantic consistency. Reviewers may also directly observe this effect from the demo video in our project website: [https://sites.google.com/view/tric-motion-iclr2026/%E9%A6%96%E9%A1%B5](https://sites.google.com/view/tric-motion-iclr2026/%E9%A6%96%E9%A1%B5).

Taken together, **both quantitative and perceptual evidence support our claim that the proposed method improves generation quality in a meaningful and practically relevant sense, especially in terms of accurate and faithful text-conditioned motion generation**.

---

### Author Response · Authors · 2025-11-29
**Response to Reviewers' Comments (Page 6)**

**[To Reviewer2-7LNK]**

**Q-B3:** Results from strong baselines are missing. It is necessary to compare the proposed method against recent, stronger approaches (_e.g._, MARDM, MotionStreamer, MotionLCM v2, SALAD) using both qualitative and quantitative evaluations.

**A-B3**: We thank the reviewer for emphasizing the importance of including stronger recent baselines. In response, we have conducted comprehensive qualitative and quantitative comparisons against MARDM [4], MotionLCM-v2 [5], and SALAD [2], and have integrated these results into the revised manuscript. In addition, we have carried out a user study to further assess perceptual motion quality and text–motion alignment from a human-evaluation perspective. These added experiments provide a more complete and rigorous comparison with recent state-of-the-art approaches.

**About MotionStreamer:** MotionStreamer [14] adopts a different motion representation and evaluates on a new evaluator. Therefore, **a direct numerical comparison is not feasible**, and we omit it here to avoid misleading or incomparable results. Nonetheless, we **include qualitative comparisons with MotionStreamer in our demo videos** for completeness (see the project website and Table 5 in the paper).

**Qualitative Comparison:**

1. Extensive side-by-side video comparisons with these methods have been added to our project website:
 [https://sites.google.com/view/tric-motion-iclr2026/%E9%A6%96%E9%A1%B5](https://sites.google.com/view/tric-motion-iclr2026/%E9%A6%96%E9%A1%B5).
2. We will also update Figure 5 in the revised manuscript to include full quantitative comparisons with more strong baselines.

**Quantitative Comparison:** The additional results of Table 1 in our paper (on the HumanML3D dataset) are summarized in Table B2 below, and for ease of comparison, the table also includes the experimental results of TriC-Motion. As shown in Table 1 and Table B2, TriC-Motion demonstrates a clear advantage in text–motion alignment. Across all three R-Precision metrics, our model consistently outperforms the strong baselines. In particular, our method (Large) achieves an R@1 of 0.612, exceeding MARDM by +0.112, MotionLCM-V2 by +0.061, and SALAD by +0.031. In addition, our method reduces MM-Dist to 2.463, improving over MARDM by −0.807, over MotionLCM-V2 by −0.302, and over SALAD by −0.186. These results demonstrate that, by leveraging its unified tri-domain modeling framework together with causal intervention, TriC-Motion achieves substantially superior text–motion consistency compared with existing state-of-the-art methods.

Table B2: Additional quantitative results on HumanML3D. The best-performing results are highlighted in bold.
| **Methods**                         | **_R@1 ↑_** | **_R@2 ↑_** | **_R@3 ↑_** | **_FID ↓_**       | **_MM-Dist ↓_**     | **_Diversity_**    |
|----------------------------------|-----------------------|-----------------------|-----------------------|-------------------|---------------------|---------------------|
| **MARDM (CVPR 2025)** [4]        | _0.500 ± 0.004_       | _0.695 ± 0.003_       | _0.795 ± 0.003_       | _0.114 ± 0.007_   | _3.270 ± 0.009_     | _-_                 |
| **MotionLCM-V2 (ECCV 2024)** [5] | _0.551 ± 0.003_       | _0.745 ± 0.002_       | _0.836 ± 0.002_       | _0.049 ± 0.003_| _2.765 ± 0.008_     | _9.584 ± 0.066_     |
| **SALAD (CVPR 2025)** [2]        | _0.581 ± 0.003_       | _0.769 ± 0.003_       | _0.857 ± 0.002_       | _0.076 ± 0.002_   | _2.649 ± 0.009_     | _9.696 ± 0.096_     |
| **GMMotion (NIPS 2025)** [15] | _0.572 ± 0.003_       | _0.761 ± 0.003_       | _0.852 ± 0.001_       | _0.086 ± 0.003_   | _2.743 ± 0.008_     | _9.792 ± 0.085_     |
| **MoMask++ (NIPS 2025)** [16]      | _0.528 ± 0.003_       | _0.718 ± 0.003_       | _0.811 ± 0.002_       | _0.072 ± 0.003_   | _2.912 ± 0.008_     | _-_                 |
| **FlashMo (NIPS 2025)** [17]       | _0.568 ± 0.005_       | _0.761 ± 0.002_       | _0.851 ± 0.003_       | **_0.029 ± 0.002_**| _2.703 ± 0.005_     | _9.601 ± 0.073_     |
| **LaMP (ICLR 2025)** [6]             | _0.557 ± 0.003_       | _0.751 ± 0.002_       | _0.843 ± 0.001_       | _0.032 ± 0.002_| _2.759 ± 0.007_     | _9.571 ± 0.069_     |
| **Ours (Base)**           | _0.607 ± 0.005_       | _0.800 ± 0.004_       | **_0.886 ± 0.004_**   | _0.347 ± 0.031_   | **_2.463 ± 0.012_** | _9.428 ± 0.085_     |
| **Ours (Large)**          | **_0.612 ± 0.006_**   | **_0.806 ± 0.005_**   | _0.885 ± 0.004_       | _0.285 ± 0.042_   | _2.465 ± 0.017_     | _9.434 ± 0.089_     |

---

### Author Response · Authors · 2025-11-29
**Response to Reviewers' Comments (Page 7)**

## Part C: About the visualizations of TriC-Motion.

**[To Reviewer1-CRLx, Reviewer2-7LNK]**

**Q-C1:** The **lack of a demo video** is a significant issue. Visualization results are necessary because quantitative metrics in text-to-motion tasks are often fragile and can misalign with human judgment. Without supplementary demo videos, it is difficult to evaluate the actual quality of the generated motions.

**A-C1:** We appreciate the reviewer’s observation that quantitative metrics in text-to-motion generation may not always align with human perceptual judgment, and we agree that visualization and demo videos are essential for a fair evaluation of motion quality. We apologize for the omission in the initial submission.

To address this concern, we have prepared a dedicated **website containing extensive qualitative results, including full demo videos, side-by-side comparisons with state-of-the-art methods, and t-SNE visualizations** : [https://sites.google.com/view/tric-motion-iclr2026/%E9%A6%96%E9%A1%B5](https://sites.google.com/view/tric-motion-iclr2026/%E9%A6%96%E9%A1%B5). It is important to note that we strictly adhere to the double-blind review principles of the ICLR 2026 conference, ensuring the anonymity of the website.

These visualizations clearly show that TriC-Motion produces more faithful text–motion alignment (accurate motion order, direction, and temporal consistency) and higher visual motion quality (more stable body topology, smoother transitions, and richer motion details) compared with representative SOTA baselines, including MARDM [4], MoMask [1], and SALAD [2].




**[To Reviewer1-CRLx, Reviewer2-7LNK]**


**Q-C2:** The paper does not include a **user study** as supplementary material to subjectively assess text-motion alignment and motion quality. These are essential for a comprehensive evaluation of the method.

**A-C2:** To further validate perceptual quality, we additionally conduct a user study following the protocol adopted by GMMotion [15] and SALAD [2].
To evaluate the perceptual quality of text-driven motion generation, we conducted a user study with 38 participants comparing our method against three representative baselines: MARDM [4], MoMask [1], and SALAD [2], which respectively represent MAR paradigms, discrete AR models, and continuous diffusion frameworks. For each method, participants were presented with 15 video examples and asked to evaluate them on two criteria: visual quality and text–motion alignment. All ratings were collected using a 5-point Likert scale ranging from 1 (poorest) to 5 (best). The results, summarized in Table C1, demonstrate that **TriC-Motion consistently outperforms all competing methods in both visual quality and text–motion alignment**.

**To facilitate better interaction with the reviewers, we provide the questionnaire used in the user study. Unlike the version distributed to participants, this questionnaire reveals which method generated the motion immediately after each evaluation (whereas participants were unaware of the method used to generate each motion throughout the experiment)**. The questionnaire strictly adheres to the double-blind review policy of ICLR 2026, and the link is as follows: [https://wj.qq.com/s2/25051648/ed79/](https://wj.qq.com/s2/25051648/ed79/).

Table C1: Score results of user study using different methods.
| **Methods**          | **_Text-Motion Alignment_** | **_Overall Quality_**  |
|-------------------|-----------------------------|-------------------------|
| **Ours**          | **_4.125 ± 0.087_**         | **_3.981 ± 0.091_**     |
| **SALAD** [2]     | _3.518 ± 0.108_             | _3.567 ± 0.099_         |
| **MARDM** [4]   | _2.970 ± 0.108_             | _3.268 ± 0.100_         |
| **MoMask** [1]    | _3.096 ± 0.112_             | _3.289 ± 0.103_         |

---

### Author Response · Authors · 2025-11-29
**Response to Reviewers' Comments (Page 8)**

**[To Reviewer3-QW9h]**

**Q-C3:** Please provide **t-SNE** or other visualization analyses that disentangle motion-irrelevant and motion-relevant information to demonstrate the effectiveness of the proposed method.

**A-C3:** We present t-SNE visualizations of six different text inputs, from the HumanML3D test dataset, processed by TriC-Motion, showing the motion-irrelevant features $F_f$, motion-relevant features $F_{cf}$, and the final features contributing to generation $F_{tde}$ obtained through causal modeling. As shown in the t-SNE visualizations on our website, for certain complex or easily confusable motion texts, their features $F_f$ overlap in the space. After being modeled by the CCMD module based on causal learning, the disentangled features $F_{tde}$ clearly demonstrate that the features corresponding to each text become separated, and features within the same category are more clustered. This indicates that **after removing the confounding information and motion-relevant factors, the model generates more accurate and less ambiguous motion features based on the text, thereby improving the consistency between generated motion and text as well as the fidelity of the motion**. We have visualized these results on our project website [https://sites.google.com/view/tric-motion-iclr2026/%E9%A6%96%E9%A1%B5](https://sites.google.com/view/tric-motion-iclr2026/%E9%A6%96%E9%A1%B5).



**[To Reviewer3-QW9h]**

**Q-C4:** The methods compared in Figure 5 in our paper are general approaches, and it is unclear whether the proposed method outperforms contemporary spatio-temporal modeling methods. If possible, qualitative comparisons (or quantitative comparisons, if allowed) should be provided against methods from papers accepted to CVPR 2025, ICCV 2025, and NeurIPS 2025.

**A-C4:** We appreciate the suggestion to compare against more recent spatio-temporal methods. Following the reviewer’s advice, we have added quantitative results for a series of contemporary methods in Table B2, and we will carefully revise Table 1 in our paper.

For quantitative comparison, our method achieves R@1 of 0.612, outperforming all contemporary baselines on HumanML3D dataset. The same advantage holds for R@2 and R@3. We also obtain the lowest MM-Dist across all compared methods, reaching 2.463, noticeably lower than strong baselines such as SALAD [2] (2.649), MotionStreamer [14] (2.765), and GMMotion [15] (2.743). These results indicate that even when compared with the latest spatio-temporal architectures, TriC-Motion provides the strongest text–motion alignment and maintains competitive motion fidelity.

For qualitative comparison, we have built a temporary project website [https://sites.google.com/view/tric-motion-iclr2026/%E9%A6%96%E9%A1%B5](https://sites.google.com/view/tric-motion-iclr2026/%E9%A6%96%E9%A1%B5) that hosts extensive visualization videos, including side-by-side comparisons against SALAD [2], MARDM [4], MotionLCM-v2 [5], and MotionStreamer [14] on the same prompts. The visualizations highlight that our method better preserves complex action order, directional cues, and motion details, supporting the quantitative improvements in R-Precision and MM-Dist. Moreover, the visual results show that our method delivers motion quality comparable to these strong baselines, while in many scenarios the substantially improved R-Precision leads to motions that more faithfully follow the textual descriptions, resulting in overall better perceptual motion quality.

---

### Author Response · Authors · 2025-11-29
**Response to Reviewers' Comments (Page 9)**

## Part D: About the experiments and ablation studies of TriC-Motion.


**[To Reviewer1-CRLx]**

**Q-D1:** The ablation study should include more combinatorial settings to better demonstrate the effectiveness of the proposed method.

**A-D1:** We thank the reviewer for this constructive suggestion. We have conducted additional ablation experiments with more combinatorial settings, and we will include them in the revised version of the paper.

**Ablations of tri-domain modeling framework:**

As shown in Table D1, we progressively add STM, HFA, and S-Fus on top of TME (i.e., TME+STM, TME+HFA, TME+STM+HFA, and TME+STM+HFA+S-Fus) to reveal how each domain-specific module contributes and how their interactions lead to the final performance. Note that when S-Fus is not used, the tri-domain features are combined via simple feature concatenation.
Adding STM or HFA individually already yields notable improvements. Incorporating STM leads to an R@1 improvement of approximately 0.1 and a reduction in MM-Dist, while introducing HFA leads to the most significant reduction in FID. STM enforces realistic joint topology, whereas HFA captures global motion trends and fine-grained dynamics through low- and high-frequency cues. Since spatial structure and frequency characteristics describe complementary aspects of human motion, combining STM and HFA further improves semantic alignment and motion fidelity, yielding additional gains across R-Precision and MM-Dist. Building on this foundation, the full framework equipped with S-Fus achieves the strongest overall performance, improving R@1 to 0.607 and further reducing MM-Dist. This highlights **the advantage of S-Fus over simple concatenation**, as its **adaptive scoring and domain-aware weighting more effectively leverages complementary cross-domain cues**.

Table D1: The performance comparison of ablation experiments on the main components of TriC-Motion on HumanML3D. Following MoGenTS [18], the motion sequence is preprocessed from a 1D temporal structure to a 2D spatial–temporal structure, forming $X \in R^{N\times M\times 12}$, denoted as _2D rep_ in the table, where N denotes the number of temporal frames and M denotes the number of joints. The best-performing results are highlighted in bold.

| **Methods**                     | **_R@1 ↑_** | **_R@2 ↑_** | **_R@3 ↑_** | **_FID ↓_**       | **_MM-Dist ↓_**     | **_Diversity_**    |
|------------------------------|------------------------|------------------------|------------------------|-------------------|---------------------|----------------------|
| **MDM (Baseline)**          | _0.491 ± 0.006_       | _0.709 ± 0.006_       | _0.815 ± 0.005_       | _0.495 ± 0.041_   | _3.04 ± 0.016_      | _9.88 ± 0.098_       |
| **TME (2D rep)**            | _0.470 ± 0.006_       | _0.671 ± 0.006_       | _0.777 ± 0.007_       | _2.110 ± 0.071_   | _3.293 ± 0.023_     | _8.016 ± 0.094_      |
| **TME+STM**                 | _0.570 ± 0.004_       | _0.771 ± 0.005_       | _0.859 ± 0.005_       | _0.611 ± 0.065_   | _2.617 ± 0.018_     | _9.157 ± 0.120_      |
| **TME+HFA**                 | _0.583 ± 0.007_       | _0.778 ± 0.004_       | _0.869 ± 0.004_       | _0.374 ± 0.015_   | _2.576 ± 0.017_     | _9.535 ± 0.072_      |
| **TME+STM+HFA**             | _0.592 ± 0.005_       | _0.780 ± 0.006_       | _0.867 ± 0.005_       | _0.383 ± 0.028_   | _2.564 ± 0.020_     | _9.679 ± 0.082_      |
| **TME+STM+HFA+S-Fus (Ours)** | **_0.607 ± 0.005_**   | **_0.800 ± 0.004_**   | **_0.886 ± 0.004_**   | **_0.347 ± 0.031_** | **_2.463 ± 0.012_** | _9.428 ± 0.085_  |


**Ablations inside the Hybrid Frequency Architecture (HFA):**

As shown in Table D2, we add w/o FFT, and w/o joint to examine the necessity of the FFT and to assess whether frequency modeling must be applied on both temporal and joint domains. Here, w/o joint means removing the joint-wise frequency branch while retaining the temporal one.

Removing the FFT branch results in a clear decline in both semantic alignment and motion fidelity, indicating that **modeling the low-frequency components within a hybrid FFT–DWT frequency space enables the network to better capture global motion structures**. Limiting HFA to the temporal domain (w/o joint) further degrades multiple metrics, demonstrating that **joint-domain frequency modeling is crucial for capturing fine-grained spatial dynamics and ensuring coherent multi-joint coordination**.

---

### Author Response · Authors · 2025-11-29
**Response to Reviewers' Comments (Page 10)**

Table D2: Performance changes of TriC-Motion on the HumanML3D dataset after removing the key components of HFA. The best-performing results are highlighted in bold.

| **Methods**             | **_R@1 ↑_** | **_R@2 ↑_** | **_R@3 ↑_** | **_FID ↓_**       | **_MM-Dist ↓_**     | **_Diversity_**    |
|-----------------------|------------------------|------------------------|------------------------|-------------------|---------------------|----------------------|
| **w/o HFA**          | _0.572 ± 0.004_       | _0.765 ± 0.004_       | _0.863 ± 0.004_       | _0.593 ± 0.052_   | _2.599 ± 0.022_     | _9.243 ± 0.083_      |
| **w/o High-band**    | _0.602 ± 0.006_       | _0.799 ± 0.005_       | _0.885 ± 0.005_       | _0.504 ± 0.048_   | _2.494 ± 0.015_     | _9.029 ± 0.080_      |
| **w/o FFT**          | _0.595 ± 0.006_       | _0.790 ± 0.006_       | _0.877 ± 0.004_       | _0.405 ± 0.050_   | _2.518 ± 0.018_     | _9.335 ± 0.095_      |
| **w/o joint**        | _0.599 ± 0.008_       | _0.793 ± 0.005_       | _0.881 ± 0.004_       | _0.418 ± 0.035_   | _2.527 ± 0.018_     | _9.188 ± 0.107_      |
| **Low+High (Ours)**  | **_0.607 ± 0.005_**   | **_0.800 ± 0.004_**   | **_0.886 ± 0.004_**   | **_0.347 ± 0.031_** | **_2.463 ± 0.012_** | _9.428 ± 0.085_  |


**Ablations on CCMD across different domains:**

As shown in Table D3, we added w/ CCMD (temp) and w/ CCMD (temp+spa) to examine how causal intervention behaves when applied to the temporal domain alone versus multiple domains jointly, and to validate whether full tri-domain causal modeling is necessary.

Applying CCMD solely to the temporal domain yields measurable improvements in text–motion alignment, with R@1 enhanced to 0.580. Extending CCMD to both temporal and spatial domains further enhances text–motion alignment and reduces MM-Dist. The full tri-domain configuration achieves the best performance, confirming that causal intervention is most effective when jointly applied across all three domains. In tri-domain modeling, substantial entangled and motion-irrelevant features inherently arise and impair generation quality. **By performing causal intervention jointly across all domains, our method more effectively isolates the intrinsic motion signals from these confounding factors**.

Table D3: The performance comparison of applying the CCMD module at different positions within the TriC-Motion framework on the HumanML3D dataset. _CCMD (post)_ indicates that the features after tri-domain fusion are processed by the CCMD module, while _temp_, _spa_, and _freq_ represent the features processed by the CCMD module in the temporal, spatial, and frequency branches, respectively. The best-performing results are highlighted in bold.

| **Methods**                             | **_R@1 ↑_** | **_R@2 ↑_** | **_R@3 ↑_** | **_FID ↓_**       | **_MM-Dist ↓_**     | **_Diversity ↑_**    |
|---------------------------------------|------------------------|------------------------|------------------------|-------------------|---------------------|----------------------|
| **w/o CCMD**                         | _0.568 ± 0.007_       | _0.767 ± 0.007_       | _0.859 ± 0.006_       | _0.561 ± 0.060_   | _2.624 ± 0.023_     | _9.187 ± 0.088_      |
| **w/ CCMD (post)**                   | _0.604 ± 0.089_       | _0.798 ± 0.007_       | _0.880 ± 0.005_       | _**0.328 ± 0.027**_   | _2.512 ± 0.018_     | _9.482 ± 0.068_      |
| **w/ CCMD (temp)**                   | _0.580 ± 0.008_       | _0.773 ± 0.008_       | _0.859 ± 0.007_       | _0.514 ± 0.049_   | _2.617 ± 0.024_     | _9.467 ± 0.092_      |
| **w/ CCMD (temp+spa)**               | _0.602 ± 0.005_       | _0.792 ± 0.006_       | _0.875 ± 0.006_       | _0.329 ± 0.034_   | _2.525 ± 0.019_     | _9.345 ± 0.094_      |
| **w/ CCMD (temp+spa+freq, Ours)**    | **_0.607 ± 0.005_**   | **_0.800 ± 0.004_**   | **_0.886 ± 0.004_**   | _0.347 ± 0.031_ | **_2.463 ± 0.012_** | **_9.428 ± 0.085_**  |

---

### Author Response · Authors · 2025-11-29
**Response to Reviewers' Comments (Page 11)**

**[To Reviewer4-p1bB]**

**Q-D2:** The architecture is computationally heavy and complex. It is important to compare the runtime, computational cost, and model size with baseline methods to provide a comprehensive evaluation.

**A-D2:** We appreciate the reviewer’s concern. To provide a clear comparison, Table D4 reports FLOPs and average inference time (AIT) measured over 100 samples **on a single NVIDIA RTX 3090 GPU** for several representative methods (MDM [3], SALAD [2], MARDM [4], MoMask [1]), and Table D5 summarizes their parameter sizes.

TriC-Motion requires 388.45 GFLOPs, only slightly higher than our baseline, and achieves an AIT of 3.8 s, which remains comparable to latent-space diffusion methods such as SALAD (3.0 s) and is substantially more efficient than MARDM (10.6 s). Importantly, the model is extremely compact, with only 13.86M parameters, significantly smaller than many strong baselines, including MoMask (44.85M), ReMoDiffuse [19] (46.97M), MLD [9] (26.38M), and especially MARDM (310.09M).

The additional computation primarily comes from explicit tri-domain modeling, which cannot be replicated by simply scaling a single-domain network. Despite incorporating temporal, spatial, and frequency branches, the overall architecture remains deliberately lightweight.

Furthermore, TriC-Motion conducts diffusion directly in the raw motion space, where multi-step denoising inherently incurs higher cost. This is a well-known limitation shared across diffusion-based motion generators. Nevertheless, the proposed tri-domain design yields clear and consistent benefits, improving R@1 to 0.612 and enhancing motion fidelity on both HumanML3D and SnapMoGen [16].

Table D4: FLOPs and average inference time (AIT) comparison across various methods.
| **Methods**                 | **_FLOPs (G)_** | **_AIT (s)_** |
|--------------------------|-----------------|----------------|
| **MDM**[3] (DDIM-50step) | _325.25_        | _1.5_          |
| **SALAD** [2]         | _233.829_       | _3.0_          |
| **MARDM** [4]           | _23519.5_       | _10.6_         |
| **MoMask** [1]     | _37.498_        | _0.4_          |
| **Ours**                 | _388.45_    | _3.8_      |

Table D5: Parameter size comparison across various methods.
| **Methods**         | **_Params (M)_** |
|------------------|-----------------|
| **MDM** [3]     | _17.88_         |
| **MLD** [9]     | _26.38_         |
| **ReMoDiffuse** [19] | _46.97_         |
| **MoMask** [1]  | _44.85_         |
| **SALAD** [2]   | _10.1_          |
| **MARDM** [4]   | _310.09_        |
| **Ours**        | _13.86_     |

It is also worth noting that **the causal module is used only during training and introduces no inference-time overhead**, and all tri-domain branches are efficient and parallelizable.

Prior works (StableMoFusion [20] and Light-T2M [21]) demonstrate that advanced samplers (e.g., DPM-Solver [22], UniPC [23]) can reduce diffusion to as few as 10 steps, yielding substantial acceleration with minimal quality degradation. These findings indicate that **TriC-Motion can be efficiently accelerated using modern sampling strategies**, and in future work we will further explore such techniques while also investigating diffusion in a more compact latent space to optimize the inference process.

---

### Author Response · Authors · 2025-11-29
**Response to Reviewers' Comments (Page 12)**

**[To Reviewer4-p1bB]**

**Q-D3:** The training stability is also worrying due to the complex architecture. The loss weighting needs more deep analysis and sensitivity test.

**A-D3:** We thank the reviewer for raising the concern regarding training stability and the weighting of the loss terms. To address this point, we conducted a comprehensive sensitivity analysis on both the perceptual loss $L_p$ and the causal loss $L_{fcf}$, as shown in Table D6 and Table D7. The results consistently demonstrate that TriC-Motion remains stable under a wide range of loss weights. Varying $L_p$ from 1 to 20 produces only minor fluctuations in R-Precision (0.597 vs 0.607) and FID (0.347 vs 0.425), without any sign of instability or performance degradation. Similarly, adjusting $L_{fcf}$ from 0.5 to 2 yields very close outcomes, indicating that **the causal intervention branch does not introduce sensitivity and instability**. These findings confirm that **our model is not sensitive to the loss weights**.

Table D6: Sensitivity analysis of the perceptual loss $L_P$. $\alpha$ and $\beta$ denote the weights of $L_{fcf}$ and $L_P$, respectively.
| **Sensitivity of $L_P$**           | **$\alpha$** | **$\beta$** | **_R@1 ↑_** | **_R@2 ↑_** | **_R@3 ↑_** | **_FID ↓_**       | **_MM-Dist ↓_**     | **_Diversity _**    |
|--------------------|-------------|-----------|------------------------|------------------------|------------------------|-------------------|---------------------|----------------------|
|   | _1.0_       | _10_      | _0.607 ± 0.005_        | _0.800 ± 0.004_        | _0.886 ± 0.004_        | _0.347 ± 0.031_   | _2.463 ± 0.012_     | _9.428 ± 0.085_      |
|                   | _1.0_       | _1_       | _0.597 ± 0.006_        | _0.787 ± 0.006_        | _0.867 ± 0.005_        | _0.416 ± 0.035_   | _2.571 ± 0.023_     | _9.917 ± 0.066_      |
|                   | _1.0_       | _20_      | _0.600 ± 0.006_        | _0.794 ± 0.005_        | _0.878 ± 0.004_        | _0.425 ± 0.037_   | _2.517 ± 0.017_     | _9.229 ± 0.074_      |


Table D7: Sensitivity analysis of the causal loss $L_{fcf}$. $\alpha$ and $\beta$ denote the weights of $L_{fcf}$ and $L_P$, respectively.

| **Sensitivity of $L_{fcf}$**    | **$\alpha$** | **$\beta$** | **_R@1 ↑_** | **_R@2 ↑_** | **_R@3 ↑_** | **_FID ↓_**       | **_MM-Dist ↓_**     | **_Diversity _**    |
|--------------------|-------------|-----------|------------------------|------------------------|------------------------|-------------------|---------------------|----------------------|
|  | _0.5_       | _10_      | _0.609 ± 0.005_        | _0.803 ± 0.004_        | _0.882 ± 0.004_        | _0.349 ± 0.032_   | _2.485 ± 0.017_     | _9.464 ± 0.071_      |
|                   | _2_         | _10_      | _0.599 ± 0.008_        | _0.793 ± 0.005_        | _0.881 ± 0.004_        | _0.418 ± 0.035_   | _2.527 ± 0.018_     | _9.188 ± 0.107_      |


## Part E: About the innovations of our method.

**[To Reviewer3-QW9h]**

**Q-E1:** What advantages does using DistilBERT for word-level and sentence-level feature encoding have compared to CLIP?

**A-E1:** HumanML3D and SnapMoGen contain long, compositional sentences with multiple verbs and modifiers (*e.g.*, “walks forward, suddenly stops, turns right, then sits”). CLIP text embeddings are optimized for global vision-language alignment and tend to compress such sequences into a single global embedding. In contrast, **DistilBERT preserves word-level structure, enabling our TIJ module to attend explicitly to key linguistic elements, such as individual verbs, body-part modifiers, directional cues, and sequential connectors**. This fine-grained semantic decomposition is essential for grounding multi-action, temporally ordered motions and yields more faithful text–motion alignment.

CLIP is tightly coupled with an image encoder through contrastive learning. This alignment is beneficial for image generation, but motion generation requires features that correlate with dynamic and procedural actions, not static appearance cues. DistilBERT, trained purely on language modeling, produces embeddings that better represent action semantics, temporal markers, and verb-phrase hierarchy.

---

### Author Response · Authors · 2025-11-29
**Response to Reviewers' Comments (Page 13)**

**[To Reviewer3-QW9h]**

**Q-E2:** The proposed TME, STM, HFA, and S-Fus are commonly used extraction and fusion strategies in the temporal–spatial–frequency domain and lack novelty.

**A-E2:** We thank the reviewer for the constructive comment. We would like to clarify that the core contribution of our work does not lie in proposing new standalone architectures for temporal, spatial, or frequency encoding. Instead, **our novelty is in the first unified framework that jointly models all three domains in a diffusion denoising architecture, and in demonstrating that their causal disentanglement and score-guided fusion leads to substantial performance gains**.

Existing works typically model these domains independently or at most jointly in pairwise combinations. For example, temporal-only modeling (MDM [3]), spatio-temporal modeling (MoGenTS [18], SALAD [2]), or frequency-only modeling (DiffusionPhase [24], FTMoMamba [25]). To our knowledge, no prior diffusion-based text-to-motion method simultaneously integrates temporal, spatial, and frequency cues, nor do existing approaches address the inter-domain redundancy and interference that inevitably emerge in multi-branch architectures.

Thus, while the individual modules (e.g., Transformer, GCN) are standard building blocks, as is common in most motion-generation architectures, the innovation lies in how they are organized, unified and fused, resulting in a new modeling paradigm that has not been explored in prior text-to-motion diffusion models.



**[To Reviewer4-p1bB]**

**Q-E3:** The main idea of modeling all three domains at the same time is less convincing and requires more intuitive explanation and theoretically grounded analysis.

**A-E3**: Thank you for this insightful comment. Below we clarify the intuition and theoretical foundation of modeling spatial, temporal, and frequency domains jointly.

The TriC-Motion framework models spatial, temporal, and frequency domains simultaneously, leveraging their complementary roles in human motion. The temporal domain ensures sequence continuity, the spatial domain encodes joint topology and physical plausibility, and the frequency domain captures global trends and fine-grained dynamics. **This tri-domain approach addresses limitations of single- or dual-domain methods, as omitting spatial modeling leads to unrealistic joint configurations, while discarding frequency modeling removes dynamic details**.

The tri-domain design is theoretically supported by a joint optimization mechanism that reduces noise accumulation and enhances motion fidelity. The Score-guided Tri-domain Fusion module balances domain-specific contributions through semantic and motion-aware scoring, while the Causality-based Counterfactual Motion Disentangler isolates motion-irrelevant noise and preserves causal factors during denoising.

Experimental validation confirms the effectiveness of this approach, with TriC-Motion achieving state-of-the-art R-Precision and competitive FID performance. These results demonstrate the necessity and benefits of jointly modeling all three domains.



## Part F: Typo in our paper.

**[To Reviewer2-7LNK]**

**Q-F1:** There is a minor typo in the introduction, where R@1 is stated as 0.573, which does not match the value presented in Table 1.

**A-F1:** We thank the reviewer for pointing this out. The value 0.573 in the introduction is a typo. The correct R@1 for our method is 0.612, consistent with Table 1. We have fixed this typo and ensured that all reported numbers are fully aligned across the paper.

---

### Author Response · Authors · 2025-11-29
**Response to Reviewers' Comments (Page 14, end page)**

The above constitutes our complete responses to the reviewers' comments. Once again, we sincerely thank you for your thorough review! If you have any further questions, please feel free to reach out at any time.

## Reference

[1] MoMask: Generative Masked Modeling of 3D Human Motions

[2] SALAD: Skeleton-aware Latent Diffusion for Text-driven Motion Generation and Editing.

[3] Human Motion Diffusion Model.

[4] Rethinking Diffusion for Text-Driven Human Motion Generation: Redundant  Representations, Evaluation, and Masked Autoregression.

[5] MotionLCM: Real-time Controllable Motion  Generation via Latent Consistency Model.

[6]  LaMP: Language-Motion Pretraining for Motion Generation, Retrieval, and Captioning.

[7] MotionPCM: Real-Time Motion Synthesis with Phased Consistency Model.

[8] CLaM: An Open-Source Library for Performance Evaluation of Text-driven Human Motion Generation.

[9] Executing your Commands via Motion Diffusion in Latent Space.

[10] MotionDiffuse: Text-Driven Human Motion Generation with Diffusion Model.

[11] T2M-GPT: Generating Human Motion from Textual Descriptions with  Discrete Representations.

[12] MotionGPT: Human Motion as a Foreign Language.

[13] Generating Diverse and Natural 3D Human Motions from Text.

[14] MotionStreamer: Streaming Motion Generation via Diffusion-based  Autoregressive Model in Causal Latent Space.

[15] Autoregressive Motion Generation with Gaussian Mixture-Guided Latent Sampling.

[16] SnapMoGen: Human Motion Generation from Expressive Texts.

[17] FlashMo: Geometric Interpolants and Frequency-Aware Sparsity for Scalable Efficient Motion Generation.

[18] MoGenTS: Motion Generation based on Spatial-Temporal Joint Modeling.

[19] ReMoDiffuse: Retrieval-Augmented Motion Diffusion Model.

[20] StableMoFusion: Towards Robust and Efficient Diffusion-based Motion Generation Framework.

[21] Light-T2M: A Lightweight and Fast Model for Text-to-motion Generation.

[22] DPM-Solver: A Fast ODE Solver for Diffusion Probabilistic Model Sampling in Around 10 Steps.

[23] UniPC: A Unified Predictor-Corrector Framework for Fast Sampling of Diffusion Models.

[24] DiffusionPhase: Motion Diffusion in Frequency Domain.

[25] FTMoMamba: Motion Generation with Frequency and Text State Space Models.

---

### Author Response · Authors · 2025-12-01
**Summary of Our Rebuttal Work**

## **Summary of Our Rebuttal Work**

For your quick reference, we have summarized all the work completed during the rebuttal period as follows:


#### **Part A: Proposed Framework, Baselines, and Evaluators**
1. **Baseline Choice and Compatibility**:
   - Clarified the architectural incompatibility of our method with discrete-token models like MoMask (CVPR 2024).
   - Demonstrated compatibility by porting our framework to SALAD (CVPR 2025), a continuous diffusion model, achieving competitive results while preserving motion quality and text consistency.

2. **Evaluator Weakness**:
   - Re-evaluated our method using the stronger CLaM evaluator (ACMMM 2024), showing substantial improvements in R-Precision, surpassing baselines like SALAD and the ground truth.

3. **Perceptual Loss Concerns**:
   - Conducted ablation studies to confirm that R-Precision improvements are not solely due to the perceptual loss $L_P$.
   - Evaluated the same trained model using the stronger CLaM evaluator, demonstrating that performance gains persist even with a different evaluation feature space.
   - Verified that the improvements originate from our tri-domain modeling and causal intervention design, rather than evaluator coupling or the use of $L_P$.


#### **Part B: Performance of TriC-Motion**
1. **R-Precision and FID Concerns**:
   - Explained that higher-than-ground-truth R-Precision reflects superior text–motion alignment, supported by qualitative comparisons, user studies, and additional experiments.
   - Addressed FID concerns by porting the method to SALAD, achieving significant FID improvements, demonstrating that our contributions are orthogonal to the baseline choice.
   - Highlighted improvements in motion fidelity and text–motion alignment, supported by user studies and qualitative comparisons.


2. **Comparison with Strong Baselines**:
   - Conducted comprehensive evaluations against recent methods (e.g., MARDM (CVPR 2025), MotionLCM-v2 (ECCV 2024), SALAD), showing consistent superiority in R-Precision and MM-Dist.


#### **Part C: Visualizations of TriC-Motion**
1. **Demo Videos**:
   - Created a dedicated project website with extensive qualitative results, including side-by-side comparisons and t-SNE visualizations, to validate motion quality and text alignment.
   - Project website: [https://sites.google.com/view/tric-motion-iclr2026/%E9%A6%96%E9%A1%B5](https://sites.google.com/view/tric-motion-iclr2026/%E9%A6%96%E9%A1%B5).

2. **User Study**:
   - Conducted a user study with 38 participants, confirming that our method outperforms baselines in both visual quality and text–motion alignment.
   - User study questionnaire for reference: [https://wj.qq.com/s2/25051648/ed79/](https://wj.qq.com/s2/25051648/ed79/).

3. **t-SNE Visualizations**:
   - Provided t-SNE results demonstrating effective disentanglement of motion-relevant and motion-irrelevant features, improving text–motion consistency.

4. **Comparison with Recent Methods**:
   - Added qualitative and quantitative comparisons with recent spatio-temporal models, highlighting our method’s superior alignment and competitive motion fidelity.


#### **Part D: Experiments and Ablation Studies**
1. **Combinatorial Ablation Settings**:
   - Conducted detailed ablation studies on tri-domain modeling, HFA module, and CCMD module, confirming their individual and combined contributions.

2. **Efficiency and Complexity**:
   - Compared runtime, FLOPs, and parameter sizes with baselines, demonstrating that TriC-Motion is lightweight and computationally efficient.

3. **Training Stability and Sensitivity**:
   - Conducted sensitivity analyses on loss weights, showing stable performance across a wide range of configurations.


#### **Part E: Innovations of the Method**
1. **DistilBERT vs. CLIP**:
   - Justified the use of DistilBERT for preserving word-level structure, enabling finer semantic grounding for multi-action motions.

2. **Novelty of Tri-Domain Modeling**:
   - Clarified that the novelty lies in the unified tri-domain modeling framework and its causal disentanglement mechanism, which has not been explored in prior diffusion-based methods.

3. **Theoretical Justification**:
   - Explained the complementary roles of spatial, temporal, and frequency domains, supported by empirical results and theoretical analysis.


#### **Part F: Typo**
1. **Typo Correction**:
   - Corrected a typo in the introduction where R@1 was misstated, ensuring consistency with Table 1.

If you have any questions, please feel free to discuss them with us. :D

---

### Meta-Review · Area_Chair_Uxpv · 2025-12-30

**Summary:**

This paper studies text-to-motion generation and received four reviews. The reviewers raised major concerns regarding the need for additional details of the proposed method, more thorough performance evaluation, additional visualization comparisons, and clarification of the novelty. The authors provided detailed responses to each concern and question. None of the reviewers engaged in the discussion during the limited discussion period. Two reviewers gave initially positive ratings, while the others gave negative scores.

The AC carefully reviewed all the submitted reviews and the rebuttal and concluded that the rebuttal adequately addressed the feedback from the reviewers with negative ratings, including ablation studies on loss terms, additional videos demonstrating the generated motions, clarification of evaluation metrics, results and comparisons with stronger baselines, and further explanation of the novelty. Overall, the paper is recommended for acceptance.

**Reviewer Concerns:**

The rebuttal addressed all the concerns and answered all the questions raised by the reviewers.

**Reviewer Scores:**

None of the reviewers participated in the discussion during the limited time period. The authors provided a detailed rebuttal addressing all questions and concerns. The two reviewers with positive ratings are likely to maintain their positive scores. The reviewers with negative ratings may change their scores to positive, as their major concerns have been addressed in the rebuttal.

---

### Decision · Program_Chairs · 2026-01-26

Accept (Poster)